# xTrimoGene: An Efficient and Scalable Representation Learner for Single-Cell RNA-Seq Data

**Jing Gong**[1][*]  **Minsheng Hao**[12][*]  **Xingyi Cheng**[1][†]  **Xin Zeng**[1]
**Chiming Liu**[1]  **Jianzhu Ma**[2]  **Xuegong Zhang**[2]  **Taifeng Wang**[1]  **Le Song**[13][†]
[1] BioMap Research      [2] Tsinghua University
[3] Mohamed bin Zayed University of Artificial Intelligence
{gongjing, minsheng_2022, xingyi, zengxin, chiming, taifeng, songle}@biomap.com,
{zhangxg, majianzhu}@tsinghua.edu.cn

## Abstract

Advances in high-throughput sequencing technology have led to significant progress in measuring gene expressions at the single-cell level. The amount of publicly available single-cell RNA-seq (scRNA-seq) data is already surpassing 50M records for humans with each record measuring 20,000 genes. This highlights the need for unsupervised representation learning to fully ingest these data, yet classical transformer architectures are prohibitive to train on such data in terms of both computation and memory. To address this challenge, we propose a novel asymmetric encoder-decoder transformer for scRNA-seq data, called xTrimoGene$^\alpha$ (or xTrimoGene for short)[4], which leverages the sparse characteristic of the data to scale up the pre-training. This scalable design of xTrimoGene reduces FLOPs by one to two orders of magnitude compared to classical transformers while maintaining high accuracy, enabling us to train the largest transformer models over the largest scRNA-seq dataset today. Our experiments also show that the performance of xTrimoGene improves as we scale up the model sizes, and it also leads to SOTA performance over various downstream tasks, such as cell type annotation, perturb-seq effect prediction, and drug combination prediction. xTrimoGene model is now available for use as a service via the following link: https://api.biomap.com/xTrimoGene/apply.

## 1   Introduction

Recently, Artificial Intelligence (AI) technology has demonstrated promising results for addressing scientific problems. This AI4Science paradigm witnessed diverse successful biological and pharmaceutical applications, including protein analysis [17, 20, 38, 35, 1], RNA modeling [4], and genomics modulation [27]. However, most existing AI models have predominantly focused on protein sequences, neglecting the growing volume of high-throughput experimental sequencing data in the form of gene expression values. Single-cell RNA sequencing (scRNA-seq) technology has transformed the field of cell biology and enabled us to understand cell-cell, cell-gene and gene-gene relations at the cellular level [16, 3]. This technique captures the expression levels of thousands of genes in parallel, facilitating the study of cellular heterogeneity [5, 19]. This unveiled information is crucial for understanding complex biological systems and disease progression [16, 3]. Integrating and

---

[*]Equal contribution. Mingsheng Hao conducted this work during his internship at BioMap.

[†]Correspondence to: xingyi@biomap.com, songle@biomap.com.

[4]xTrimoGene is a member of BioMap's large-scale AI engine "xTrimo" series. "$\alpha$" denotes the academic version, to distinguish it from the commercial product xTrimoGene.

37th Conference on Neural Information Processing Systems (NeurIPS 2023).

modeling such large-scale scRNA-seq data can reveal rich cellular information and benefit various biological task learning.

Representation learning from scRNA-seq data [9] has been an active area of research in past decades. For example, scVAE [11] and scVI [21] apply a variational autoencoder framework to derive low-dimensional cell embeddings, cscGAN [24] uses a Generative Adversarial Network (GAN) architecture to generate cell-type specific expression profiles, and SAVER-X [32] is capable of removing batch effects across datasets. Despite the success of these customized algorithms, they tend to be computationally inefficient and labor-intensive. This prompts us to explore a general-purpose model that first learns underlying knowledge from scRNA-seq data and generalizes it to different tasks in a unified manner. We draw inspiration from the pre-training and fine-tuning paradigm in Natural Language Processing (NLP), which has shown great success in improving various downstream NLP task performance [29, 12, 14]. In light of these findings, we aim to investigate the potential of applying similar approaches to representation learning in scRNA-seq data.

The first published pre-trained model for single-cell data is scBERT, which uses a low-rank transformer [36] to analyze the scRNA data. It learns the cellular representation by randomly masking a percent of non-zero gene expression values and tries to recover them. scBERT has achieved state-of-the-art results for cell-type annotation tasks. The study shows the potential of a pre-training strategy for single-cell biology research. However, scBERT has certain limitations in fully utilizing scRNA-seq data properties. These limitations include:

(1) Scalability. The large number of genes (almost 20,000) and the sparsity of scRNA-seq data, with nearly 90% of values being zero, lead to many redundant computations (e.g., self-attentions between zero tokens). It required approximately $2.65 \times 10^{19}$ FLOPs to train 5 million samples over 5 epochs, which equals almost 20 days of training on an A100 GPU for only an 8.9 million parameter scBERT model. (2) Limited resolution for expression values. scBERT rounds the gene expression values into integer values, which limits the model's ability to distinguish closeness and similarity between gene expression values. For instance, two close values could be mapped to separate embeddings (e.g., 1.99 and 2.01 are mapped to 1 and 2), and two distant values could be mapped to identical embeddings (e.g., 1.99 and 1.01 are mapped to 1). The strategy leads to a loss of resolution and introduces bias during model training, resulting in sub-optimal performance.

To address the challenges associated with scRNA-seq data modeling and consider the unique nature of this data (as discussed in Section 2), we present a novel and efficient framework, xTrimoGene, for pre-training large-scale scRNA-seq data. Our framework makes the following key contributions:

(1) We design an asymmetrical encoder-decoder architecture to guide the pre-training process, which enables us to learn a high-capacity model for single-cell RNA-seq data. Our model achieves an improvement in the speed of pre-training of over 3 times compared to previous encoder-only models.

(2) We illustrate that the efficiency and scalability of our model allow us to train the largest single-cell pre-trained model to date, with approximately 100 million parameters for the xTrimoGene-100M model, using a curated scRNA-seq dataset of approximately 50 billion effective gene tokens.

(3) The pre-trained model xTrimoGene achieved remarkable results in multiple downstream tasks, including cell type annotation, perturbation prediction and synergistic drug combination prediction.

## 2   Characteristics of Single-Cell RNA-seq Data

scRNA-seq generates a large, sparse expression matrix, where each row represents a cell (sample) and each column a gene (feature). This dataset presents several challenges and requires a specialized architecture to effectively model the data.

First, approximately 20,000 genes (columns) are shared across cells. Unlike the corpus in NLP, the genes can be arbitrarily reordered. The relation between genes depends on biological pathways rather than local contexts, where the latter shapes spatial information in Computer Vision (CV) images. Though one can roughly regard each cell (row) as a sentence or an image patch, the 20,000 genes is a vast number compared to the typical sequence length, which is mostly a few hundred and no more than a few thousand [29, 12]. Thus directly applying existing transformer architecture will not work.

Second, scRNA-seq matrices are highly sparse (90% zero in a typical dataset [15, 7]). The abundance level of RNA for each gene is measured by counting the unique molecular identifier (UMI) reads in

scRNA-seq experiments [16, 3]. However, many genes exhibit low UMI counts due to limited probing efficiency. Therefore, treating scRNA-seq data as an image and utilizing a convolutional neural network to extract features is not feasible, as it introduces a huge number of redundant computations for sparse positions.

Third, the normalized gene expression values in scRNA-seq data are continuous scalars, which typically indicate similar gene activity when they have similar values. To transform these scalars into high-dimensional tokens in the data matrix, a representation that preserves the continuous semantics is needed. Manually discretizing the gene expression values is challenging as non-optimal discretization thresholds will bias category assignment. A learned discretization approach or learnable representation, such as the one proposed in [10], is ideal for preserving the continuous semantics of the gene expression values.

Taking into the above three major features, we design a new architecture as described in the next section.

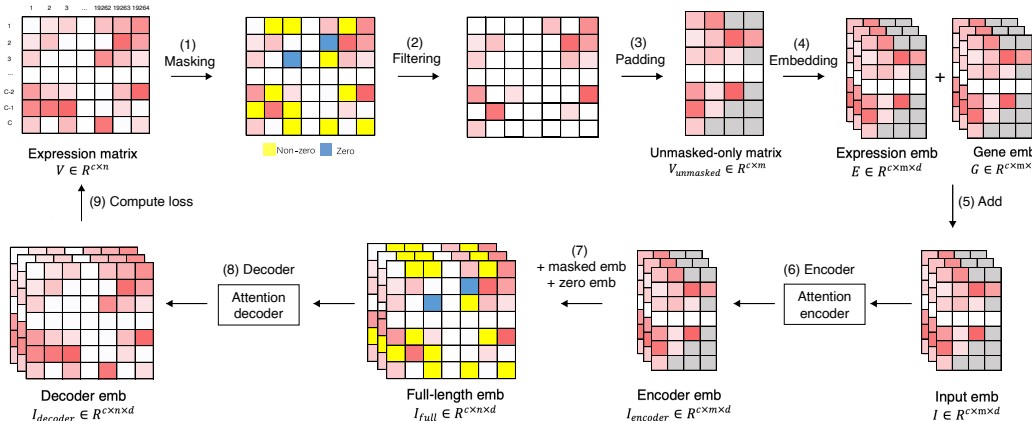

Figure 1: The xTrimoGene Framework: (1) Random positions (including both zero and non-zero values) are masked for prediction. (2) Masked and zero-valued positions are filtered out. (3) Remaining unmasked positions are aligned with padding tokens (grey) to ensure maximum length consistency within a batch. (4) Gene expression values and gene embeddings are separately projected into embeddings. (5) These two embeddings are element-wise added. (6) The resulting input is fed into the encoder. (7) The intermediate encoder embedding is combined with embeddings for masked positions and zero embeddings. (8) This combined representation is then fed into the decoder. (9) Decoder embedding is projected to model output with an MLP layer. The MSE loss is calculated between the model output and ground truth values for the masked positions.

## 3    xTrimoGene Architecture

xTrimoGene is a highly efficient framework for pre-training large-scale single-cell RNA-seq data (illustrated in Figure 1). The training process is based on a regression-masked task, aimed at accurately recovering masked values in the expression matrix. Notably, a specific optimized asymmetrical encoder-decoder framework is employed to accelerate the learning of sparse matrices. This is achieved by feeding only the unmasked non-zero positions (less than 10% of the full length) into the encoder, while the largely masked and zero positions are input into a lightweight decoder with a reduced number of layers and attention heads. In addition, a novel auto-discretization strategy is introduced to project continuous expression values into a latent embedding space. Instead of rounding to the nearest integer, values are directly mapped to the latent space allowing for the representation of closely related values. The xTrimoGene framework consists of the following components:

**Masking**: A portion of the normalized gene expression matrix $V$ is masked for prediction, including both zero and non-zero positions. $c$ denotes cell sample size, and $n$ denotes gene number (19,264 in our setting, see App. 1 for data collection and processing).

**Filtering**: The masked and zero-valued embeddings are filtered out, yielding a variable-length sequence of valuable information that is prepared for encoding.

**Padding**: The remaining unmasked positions are aligned with padding tokens, resulting in a much smaller unmasked-only matrix $V_{unmasked}$. $m$ denotes the maximum length of the unmasked sample. We include a scheme to illustrate the processing flow (see App. 2).

**Embedding**: Expression value and gene embeddings are separately projected. $d$ denotes the dimension of the embedding. The expression embedding is calculated through an auto-discretization mapping. The gene embedding is retrieved from a randomly initialized lookup table.

**Combining Expression and Gene Embeddings**: The expression and gene embeddings ($E$ and $G$) are element-wise added to form the input embedding, which is then fed into the encoder of the model.

**Encoding**: The sum of the embeddings is input into the encoder, which implements self-attention mechanisms using a Transformer-like architecture.

**Extending masked and zero embeddings**: The intermediate encoder embedding $I_{encoder}$ is combined with embeddings for masked and zero-value positions.

**Decoding**: The combined embeddings are processed by the decoder, utilizing self-attention mechanisms instead of the typical casual attention used in NLP decoders.

**Loss Computation**: Decoder embedding is projected to model output with an MLP layer. The mean squared error (MSE) loss is computed between the predicted masked values from the model and their corresponding ground truth values.

### 3.1 Encoder

The scRNA-seq data is characterized by its high sparsity, with cell information largely concentrated in the non-zero expression values. Thus, the encoder is designed to focus only on the non-zero part of the unmasked matrix, $V_{unmasked}$. The encoder is based on a traditional multi-head attention transformer and takes the combination of value embedding, $E$, and gene embedding, $G$, as its input, $I \in \mathbb{R}^{c \times m \times d}$. The value and gene embeddings are similar to the word and positional embeddings in natural language modeling, respectively. The value embedding, $E$, is generated using the auto-discretization strategy discussed previously, while the gene embedding, $G$, is retrieved from a function $f_L$ that maps the gene symbols into the embedded vocabulary.

$$E = \text{Autobin}(V_{unmasked} \odot M_{nonzero}), G = f_L(genes), I = E + G \tag{1}$$

Then the encoder processes the input embeddings $I$ and generates the high-level gene representations $I_{encoder} \in \mathbb{R}^{b \times m \times d}$ via the multi-head attention mechanism.

$$I_{encoder} = \text{Trm}(f_Q(I), f_K(I), f_V(I)) \tag{2}$$

where $f_Q$, $f_K$, $f_V$ are the project functions. Trm denotes the Transformer block.

It is worth emphasizing that our encoder only operates on a subset of genes, reducing the length of the processed sequence to 1/10 of the original. This allows the full-length transformer to be used without any computational approximations.

### 3.2 Decoder

Unlike the encoder which focuses on the main information (non-zero expression values) in the cells, the decoder in the system performs full-length feature abstraction and extraction. The input to the decoder, $I_{full}$, comprises three token types: the output from the encoder, $I_{encoder}$, the genes with zero expression embs $I_{zero}$, and the mask token embs $I_{masked}$. Out of these tokens, genes with zero expression make up 90% of all tokens. The gene embeddings are concatenated with all of these tokens to provide the decoder with gene-specific information for the corresponding mask tokens, followed by a full connection layer.

$$I_{full} = W_p(I_{encoder} \oplus I_{zero} \oplus I_{masked}) + b_p \tag{3}$$

where $\oplus$ represents the concatenation operation, and $W_p$ and $b_p$ are learnable parameters that project the decoder's embedding size.

The decoder in the framework is optimized for long-sequence attention calculations and employs the Performer architecture as its backbone. The decoder transforms the input $I_{full}$ into final gene-level embeddings, $I_{decoder} \in \mathbb{R}^{b \times n \times d}$, and predicts the masked values through a shared linear layer, $W \in \mathbb{R}^{d \times 1}$, applied across all genes. The operations are expressed as follows:

$$I_{decoder} = \mathrm{Trm}((f_Q(I_{full}), f_K(I_{full}), f_V(I_{full})), \tilde{V} = I_{decoder} \cdot W \qquad (4)$$

The decoder has a smaller model size compared to the encoder, with a smaller embedding size, fewer attention layers, and fewer attention heads. For instance, in the largest model configuration, the layer depth ratio between the encoder and decoder is 2:1 and the head number ratio is 1.5:1 (see App. Table 2). Similarly, the principle of asymmetric encoder-decoder design has been proven powerful in masked autoencoders (MAE) [13], which is tailored for CV data pre-training. Unlike MAE, xTrimoGene utilizes the biased masking strategy to avoid the learning process being dominated by zero tokens. Though the scRNA-seq data is distinct from images, our results show that the performance gains of xTrimoGene are comparable to those of MAE, with more efficient training and better downstream task performance.

### 3.3 Auto-discretization strategy

Our aim is to transform an expression value $v$ into a hidden embedding, denoted as $e$. The transformation is achieved using an auto-discretization block. This auto-discretization process involves a random look-up table $T$ defined in $\mathbb{R}^{d \times b}$. In this representation, $d$ refers to the embedding dimension, while $b$ is the number of bins with a default value of 100. The transformation starts by applying a linear layer to the expression value, given by $v_1 = v \cdot w_1$, where $w_1$ represents the weight vector. The resulting $v_1$ is then subjected to a leaky ReLU activation, resulting in $v_2 = \mathrm{Leaky\_ReLU}(v_1)$. Subsequently, a cross-layer projection is applied, represented by $v_3 = w_2 \cdot v_2 + \alpha \cdot v_2$. Here, $w_2$ denotes the weight vector, and $\alpha$ is a scaling mixture factor. Next, the bin weights of $v_3$ are normalized using the softmax function, resulting in $v_4 = \mathrm{softmax}(v_3)$. Finally, the transformed value is represented as a weighted combination of individual embeddings from the look-up table, given by $e = T \cdot v_4$. It's important to note that the weights in this combination serve as learnable parameters.

To validate the effectiveness of the expression value projection, we conducted an analysis of viewing the weight distribution pattern for continuous values. Our results showed that the normalized weight distribution of the close values exhibited smooth transitions and that of the distant values being clearly distinguishable (App. section 3 Figure 1). This supports the conclusion that the auto-discretization strategy effectively represents continuous values with high resolution while preserving relatively rich meaning.

We also compared the performance of the proposed auto-discretization strategy with three other discretization methods: (1) Round bin with zero, in which values are rounded to the nearest integer, and zeros are kept as it is, (2) Up bin without zero. Values greater than zero are converted to the nearest ceiling integer, while zero is represented as individual 0. (3) Equal bin. All the values fall into a fixed percentage interval, which is calculated by value distribution and frequency. We evaluated the different strategies on a standard cell clustering task (see App. 4) and found that the proposed auto-discretization strategy outperformed the others (as shown in Figure2 A), demonstrating the importance of high-resolution projections in handling expression values.

## 4 Training Strategy

We now explain the strategy used to train the asymmetric encoder-decoder transformer. Pre-trained task and masking strategy are outlined, see App. 5 for acceleration strategy.

### 4.1 Regression masked task

The traditional masked language task is a multi-class classification problem, where the predicting target is a single token with limited, naturally distinct categories. In contrast, the normalized gene expression value is a continuous scalar. To fit the data property, we modify the pre-trained learning

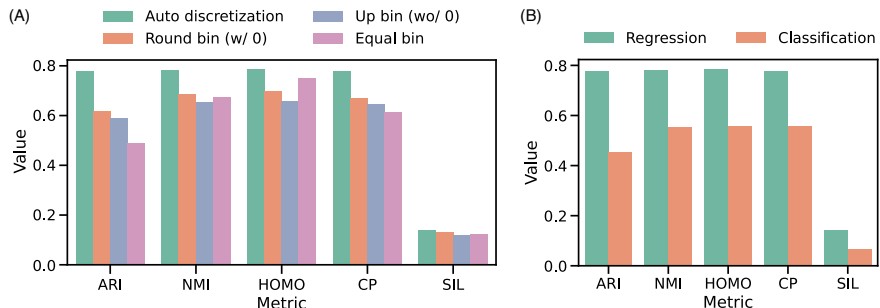

Figure 2: Pre-training strategy ablation study. (A) Performance comparison between auto discretization strategy and other binning methods for expression value projection. The cell clustering task is evaluated and five metrics are displayed. ARI for Adjusted Rand index, NMI for Normalized Mutual Information, HOMO for Homogeneity, CP for Completeness and SIL for Silhouette Coefficient. (B) Performance of pre-trained models with different task modes, including regression and classification settings. The cell clustering task is evaluated. See the main text for details.

objective to a regression task, aimed at recovering the absolute value of the masked positions. The loss function employed is the MSE between the ground truth and the predicted values:

$$\text{Loss} = \frac{1}{(n-m) * c} \sum (V_{i,j} - \tilde{V}_{i,j})^2 \tag{5}$$

where $n$ represents the number of all genes, $m$ represents the maximum length of the unmasked positions in a sample, and $c$ represents the number of cells. To evaluate the efficacy of this modification, we compared the regression setting with the classification setting on the cell clustering task. The results indicate that the regression model outperforms the classification model (Figure 2B), providing evidence of the benefits of learning a more fitted representation.

## 4.2 Masking strategy

We mask both non-zeros and zeros positions though the scRNA-seq expression matrix is highly sparse (where zero percentage is usually over 90%). As the zero positions percentage is much higher than non-zero positions, the masked ratio can't be the same for the two types. Otherwise, the model tends to predict all zeros and still obtains a low error level. We propose to mask an almost equal number of positions for zero and non-zeros positions (see App. section 6 Table 1). The setting enforces the model to learn embeddings for all values and not to be dominated by zero representation. We found zero values supervision is necessary to boost the performance (App. Figure 2), which demonstrates that some zeros represent the true extremely low expression level. This type of zeros is informative to illustrate how the gene abundant behaves inside the cell.

The recovery of masked tokens in NLP is challenging due to the fact that word comprehension relies heavily on long-range interactions rather than local context. Accurate inference of the missing tokens can be achieved at low masking ratios (15%) where the information in the entire sentence is still relatively redundant and encoded by the unmasked tokens. We investigated the density of information needed for the scRNA-seq regression task by training models with different masking ratios (for non-zero values, the ratio was set 10 times higher than for zero values) ranging from 15% to 90% with a 15% interval. The models were then evaluated on the cell clustering task, with the results showing that performance improved first and then degraded as the masking ratio increased. When the masking ratio was close to 30%, the majority of metrics reached a peak (App. Figure 3). We also found current biased masking is optimal (App. Figure 4) and the percentage of [MASK] tokens agrees well with NLP tasks (App. Figure 5). These results suggest that the scRNA-seq expression vector contains more redundant information than a sentence and highlight the role of hidden regulations between genes in constraining the inference of expression values.

# 5 Experiments

Next we will explain our experimental settings and results. The dataset description can be referred to App. 1.

## 5.1 Computational efficiency

We quantitatively compared the training cost of xTrimoGene with other two encoder-only models, including full-length attention Transformer and kernel-based approximation Performer (scBERT). For an apple-to-apple comparison, three models are set to approximately 10 million trainable parameters and trained on 5 million samples over 5 epochs. We calculated the corresponding FLOPs, where matrix multiplication operations are considered only. We observed that total FLOPs for Performer (scBERT) decreased to 10% of native Transformer (see Table 1). Notably, xTrimoGenes runs 3-times faster than Performer. The results validate the efficiency of xTrimoGene, which is readily adapted for large-scale data pre-training.

Table 1: Computational efficiency comparison between different algorithms. The resource column is normalized by the Transformer row.

| Model name | Parameter (M) | Forward + backward (FLOPs/sample) | Total train (FLOPs) | Resource |
|---|---|---|---|---|
| Transformer | 11.3 | 9.86E+12 | 2.46E+20 | 100% |
| Performer | 8.9 | 1.06E+12 | 2.65E+19 | 10.8% |
| xTrimoGene | 9.8 | 3.35E+11 | 8.38E+18 | 3.4% |

## 5.2 Scalability

The Deep Learning community has shown significant interest in the scalability of proposed models [18, 2]. Vanilla Transformer models are challenging to scale due to their computational time and resource requirements, which increase quadratically with model size. Varieties of attention mechanisms have been proposed to accelerate training speed, a critical factor for model scaling.

To test the scale-up ability of xTrimoGene, we pre-trained three models across multiple compute regions and scales (e.g., from 3M to 100M parameters). The detailed hyperparameter setting is displayed in the App. Table 2. The training curve clearly shows all models are steadily down to a lower loss when training steps increase (App. Figure 6). More importantly, the xTrimoGene-100M model obtains a significant improvement over the xTrimoGene-10M model, which is also superior to the xTrimoGene-3M model. The tendency is consistent across different data sizes. The results suggest xTrimoGene framework is robust to scale-up, making it possible and convenient to pre-train larger models with more data.

## 5.3 Robustness on high sparse data

scRNA-seq data often exhibit varying levels of sparsity, thus it's necessary to assess whether xTrimoGene is robust in handling different sparse data. To verify the robustness, we divided the test samples into subgroups based on cell type and calculated the sparsity level (i.e., percentage of zero values in the expression matrix) and Pearson correlation coefficient between the predicted and actual values. Our results reveal that the correlation gradually decreases as the sparsity level increases, as expected (Figure 3A). However, the correlation remains above 0.8 even when the sparsity level reaches 96% (Figure 3A), indicating the robustness of xTrimoGene. We also compared xTrimoGene's performance with Performer and found that xTrimoGene consistently achieves a higher correlation across most subgroups (Figure 3B). These findings demonstrate that xTrimoGene is robust in handling highly sparse data and outperforms encoder-only architectures.

The performance of the encoder-decoder and encoder-only architectures have been comparatively analyzed in the NLP domain, with the former demonstrating effectiveness in language comprehension and the latter in context generation. Apart from comparison on masked value recovery, we further evaluated xTrimoGene against encoder-only Performer on the cell clustering task. The results

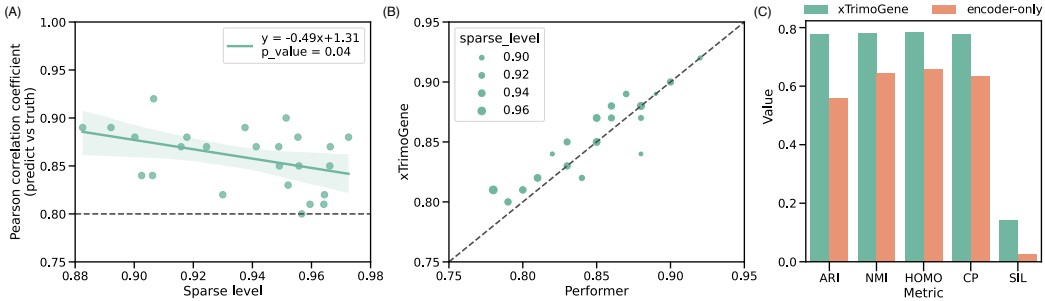

Figure 3: Comparison of performance for different sparse level data. (A) xTrimoGene performance for recovering masked values at different sparse levels. Each dot represents a subset defined by cell type. Sparse level is calculated as the ratio between zero value percentages. Pearson correlation coefficient metric is calculated on masked positions. (B) Performance comparison of xTrimoGene and Performer while recovering masked values at different sparse levels. Dot has the same meaning as (A) but the dot size is proportional to the sparse level. Both the x and y axis denotes the Pearson correlation coefficient metric for a particular algorithm. (C) Comparison of performance for xTrimoGene framework and encoder-only framework. Cell clustering task is evaluated.

demonstrate that xTrimoGene achieves superior performance, reaffirming its proficiency in latent embedding extraction (Figure 3C).

## 5.4 Evaluation on downstream tasks

Currently, multiple tasks have been established to evaluate different models, including well-defined cell type annotation and recently developed perturbation response prediction tasks. We first assessed the performance of xTrimoGene on these single-cell tasks. Additionally, we explored the potential application on bulk RNA-sequencing data, with a focus on synergistic drug combination prediction.

### 5.4.1 Cell type annotation

First, we evaluated xTrimoGene's performance on cell type annotation task with Zheng68K [39] and Segerstolpe [31] dataset, which has been widely benchmarked. We compared the xTrimoGene against other several methods, including scBERT [36], ACTINN [23], Scanpy [34], CellTypist [6], scVI [21] and singleCellNet [37]. For the xTrimoGene model, we added a max-pooling layer and a linear layer to predict cell type labels with fine-tuning mode (see App. 8.1). For other methods, we followed their instruction with the default parameter setting. We observed that xTrimoGene achieves a high Precision and F1 score, surpassing all the other methods (Table 2). The results indicated that xTrimoGene learns a well-represented cellular embedding (visualized in App. Figure 7) by simply aggregating contextual gene embedding.

Table 2: The cell annotation performance on the Zheng68K and Segerstolpe dataset. xTrimoGene is evaluated with 100M parameter model.

| Method Name | Zheng68K | | Segerstolpe | |
| --- | --- | --- | --- | --- |
| | Precision | F1 score | Precision | F1 score |
| xTrimoGene | $0.7335 \pm 0.0226$ | **0.7354** $\pm 0.0189$ | **0.8112** $\pm 0.0009$ | **0.8140** $\pm 0.0008$ |
| scBERT | $0.7029 \pm 0.0115$ | $0.6695 \pm 0.0077$ | $0.6818 \pm 0.0736$ | $0.6703 \pm 0.0653$ |
| ACTINN | $0.6720 \pm 0.0021$ | $0.6486 \pm 0.0041$ | $0.7545 \pm 0.0018$ | $0.7219 \pm 0.0073$ |
| Scanpy | $0.6111 \pm 0.0017$ | $0.5474 \pm 0.0085$ | $0.6274 \pm 0.0000$ | $0.5398 \pm 0.0000$ |
| CellTypist | **0.7454** $\pm 0.0009$ | $0.7151 \pm 0.0038$ | $0.7923 \pm 0.0003$ | $0.8117 \pm 0.0001$ |
| scVI | $0.4883 \pm 0.0005$ | $0.4843 \pm 0.0008$ | $0.5101 \pm 0.0022$ | $0.5208 \pm 0.0016$ |
| singleCellNet | $0.6452 \pm 0.0013$ | $0.5982 \pm 0.0027$ | $0.7551 \pm 0.0096$ | $0.8055 \pm 0.0076$ |

### 5.4.2 Perturbation response prediction

Recently, perturb-seq technology was established to screen gene expression response given pooled perturbations at single-cell level [8]. Several algorithms have also been developed to predict perturbation effects [30, 22] at the single cell level, i.e., what is the expression value of genes after perturbation? We compared the native GEARS[30] model with and without incorporating embeddings from xTrimoGene.

The normal state (before perturbation) gene expression profile is fed into xTrimoGene and we obtained the context embedding, which replaces raw expression value input in the GEARS model (App. 8.2). All the other settings remain unchanged. The evaluated dataset (Norman et al. [26]) contains both single and double gene perturbation and we thus assess the performance across different perturbation levels. As shown in Figure 4A, GEARS with xTrimoGene embedding scores a lower MSE (decreased 14.8%) for top20 differential expressed genes across all perturbation scenarios. Notably, the tendency is consistent across different perturbation levels, regardless the perturbed target is seen or not. We also compared against the scBERT embedding and observed a similar trend, where xTrimoGene achieves better results (App. Table 3). The results demonstrated that the pre-training strategy empowers xTrimoGene to capture constraints under various circumstances, including post-perturbations. The application further proved the efficacy and potential of xTrimoGene to boost scRNA-seq based tasks.

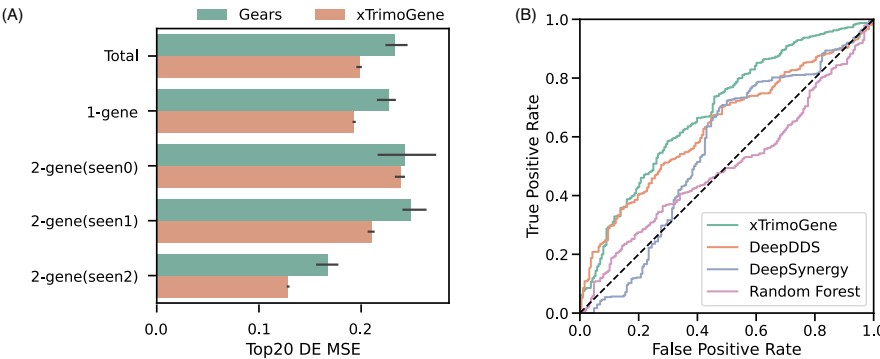

Figure 4: (A) The MSE of the top 20 deferentially expressed (DE) genes given by different models on perturbation response prediction. The top 20 DE genes are calculated between the before and post-perturbation expression profiles. "Total" denotes evaluating all test perturbation sets. "1-gene" denotes evaluation on the single gene perturbation subset, where the perturbed target is not seen in the training set. "2-gene" represents the sub-test set for perturbing two genes simultaneously. "seen0", "seen1" and "seen2" denotes zero, one or two perturbed targets are not seen in the training set, respectively. The black line denotes a 95% confidence interval. (B) ROC curve of different models on drug combination synergy prediction task. xTrimoGene denotes replacing the raw expression profile with context embeddings in the DeepDDS framework and others remain unchanged. Refer to App. 8.3 for more details.

### 5.4.3 Synergistic drug combinations prediction

The drug synergistic task evaluates how patients or cells respond to a drug combination intervention [25]. However, the generated wet-lab experimental data only covers a tiny search space of possible drug combinations. Multiple models have been proposed to accelerate predicting the synergistic landscape of drugs [28, 33]. For instance, DeepDDS integrates genomic expression profiles and drug chemical information, greatly improving the prediction performance. We further explored whether xTrimoGene is able to generate good latent embedding for this bulk expression data.

Similar to the perturbation prediction test, we adapted xTrimoGene to DeepDDS with the intermediate context embedding (see App. 8.3). We also included DeepSynergy and Random Forest for comparison. As illustrated in Figure 4B, utilizing embedding from the xTrimoGene model outperforms all the other models. The result proved that xTrimoGene can accurately capture cell-level representation, even for bulk sequencing data. This also opens the avenue for xTrimoGene to be applied across other biological modeling tasks, especially where bulk-level transcriptome data is available.

# 6 Conclusion

xTrimoGene is a new, efficient framework for learning scRNA-seq data. It proposes an asymmetric encoder-decoder framework that takes advantage of the sparse gene expression matrix and establishes the projection strategy of continuous values with a higher resolution. The results show that xTrimoGene is scalable and performs well on tasks like cell type annotation, perturbation response prediction, and synergistic drug combination prediction. The experiments demonstrate the efficacy of pre-training in single-cell biology. xTrimoGene is potentially adapted to other types of cell modeling analysis, including rare cell detection (App. 8.4), batch effect removal and regulatory network construction.

Certain limitations exist for xTrimoGene and further work is desired to advance the design. At present, xTrimoGene major utilizes gene expression values during the pre-training stage, overlooking varieties of other related meta-information like sample condition (health/disease), cell type, tissue type, sequencing platform, etc. These rich annotations are biologically meaningful and highly correlated with the expression pattern within a cell. The memory consumption for inference with the xTrimoGene-100M model is approximately 50GB, whose hardware requirement (Nvidia A100 80G GPU) is beyond some academic labs, thus computational or memory-efficient engineering techniques tend to advance the model pre-training and application.

xTrimoGene has been integrated into BioMap's single-cell analysis platform, functioning as a fundamental and essential model (as depicted in the App. Figure 9). The pre-trained model services have been publicly available. In the future, with the increase of data, larger pre-trained models are expected to drive more advancements in various downstream task learning.

## Author Contributions and Acknowledgments

Le Song led the project by designing its scope, conceptualizing ideas, integrating resources, and making decisions on techniques. Xingyi Cheng played a key role in the development of the xTrimoGene framework, auto-discretization and unsupervised objectives, contributing concrete ideas and pseudocode, along with code review. Jing Gong and Mingsheng Hao (Research Intern at BioMap) were primarily responsible for conducting pre-training and downstream experiments, serving as the first authors of the paper. Their work covered areas such as model scaling, cell type annotation, perturbation response prediction, and synergistic drug combinations prediction. Xin Zeng made significant contributions to the code of the xTrimoGene framework, worked with an early performer version, and conducted initial downstream experiments. Chiming Liu oversaw the engineering aspects of the project, including the implementation of the data pipeline and FLOPs computation. Jianzhu Ma, Xuegong Zhang, Taifeng Wang, and Le Song conceived the project and provided invaluable guidance for the project and contributed their expertise in computational biology knowledge. Taifeng Wang also played a pivotal role in pushing for the model's service implementation. Finally, Jing Gong, Mingsheng Hao, Xingyi Cheng, and Le Song collectively contributed to writing this paper.

In addition, we would like to express our gratitude to the individuals at BioMap who have made contributions to our project. Chenrui Xu and Yucheng Guo played roles in data preprocessing and integration. Zhaoren He's expertise in data analysis and application greatly enhanced our work, and we deeply appreciate his contributions.

This work was supported by the Ministry of Science and Technology of the People's Republic of China (2022YFF1203004), the Beijing Municipal Science & Technology Commission and the Administrative Commission of Zhongguancun Science Park(Z221100003522022). And this work was also funded by BioMap.

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
