# [Appendix materials]

# xTrimoGene: An Efficient and Scalable Representation Learner for Single-Cell RNA-Seq Data

**Jing Gong**[1][*]  **Minsheng Hao**[12][*]  **Xingyi Cheng**[1][†]  **Xin Zeng**[1]
**Chiming Liu**[1]  **JianZhu Ma**[2]  **Xuegong Zhang**[2]  **Taifeng Wang**[1]  **Le Song**[13][†]

[1] BioMap Research, California, USA
[2] Tsinghua University, Beijing, China
[3] Mohamed bin Zayed University of Artificial Intelligence, Abu Dhabi, UAE

{gongjing, minsheng_2022, xingyi, zengxin, chiming, taifeng, songle}@biomap.com,
{zhangxg, majianzhu}@tsinghua.edu.cn

## 1  scRNA-seq data collection and processing

Recently, scRNA-seq data is rapidly accumulated and mostly has been uploaded to the Gene Expression Omnibus (GEO) repository (https://www.ncbi.nlm.nih.gov/geo/). We collected data from GEO and processed data with a unified pipeline.

**Downloading data and preparing raw count matrix.** We first search scRNA-seq-related data sets in GEO with multiple keywords, including "scRNA-seq", "single-cell RNA-seq", and "single-cell RNA-seq sequencing". The search processes return a list of GSE ID from different studies. After removing the duplicated GSE ID, we downloaded the particular expression or count matrix. Most of the samples provide a raw count matrix. For samples with normalized expression matrices, we converted the matrix back to a count matrix. The conversion strategy is as follows: the minimal non-zero value in the whole normalized matrix is thought to have raw count 1, then all the other normalized values can be converted by scaling to this minimum value.

**Matrix mapping to the reference gene list.** After preparing all the count matrices, we mapped the matrix to our reference gene list. We downloaded the human protein-coding gene list (about 19,000) from the HUGO Gene Nomenclature Committee (HGNC, https://www.genenames.org/download/archive/), plus common mitochondrial genes, jointly constitute our full reference list ($n = 19,264$). For each count matrix, values of those genes not mapped in the reference list are filled with zero.

**Quality control.** To filter low-quality samples, we only keep samples with over 200 genes expressed (i.e., expression vector with non-zero value count > 200) for subsequent training and analysis.

**Normalization.** We followed the standard process in Scanpy (https://scanpy-tutorials.readthedocs.io/en/latest/pbmc3k.html) [7] to obtain the normalized expression value. There are two steps: (1) for each sample normalize the library size to 10,000. (2) scale the values into a log space.

In summary, all the scRNA-seq data are collected from the Gene Expression Omnibus (GEO) repository with a keyword searching and data retrieval process. Then the downloaded count matrices are processed with a unified pipeline, including reference gene list mapping, quality control and normalization. In total, we curated about 5 million scRNA-seq data for training. The full data set is randomly split into train, validation and test sets with a ratio of 96:2:2.

---

[*]Equal contribution. Mingsheng Hao conducted this work during his internship at BioMap.

[†]Correspondence to: xingyi@biomap.com, songle@biomap.com.

37th Conference on Neural Information Processing Systems (NeurIPS 2023).

## 2 Algorithm workflow

We take the below example to demonstrate the processing flow.

Assume the normalized expression value matrix has 2 cells (with 10 genes) as below:

|    | G1  | G2  | G3 | G4  | G5  | G6  | G7  | G8 | G9  | G10 |
|----|-----|-----|----|-----|-----|-----|-----|----|-----|-----|
| C1 | 0.3 | 2.1 | 0  | 4.5 | 0   | 7.3 | 8.9 | 0  | 3.4 | 2.5 |
| C2 | 1.1 | 0   | 0  | 3.4 | 2.3 | 0.7 | 0   | 0  | 2.9 | 0   |

First, we masked a portion of values (including zero and non-zero) and the generated matrix is as:

|    | G1  | G2  | G3  | G4  | G5  | G6  | G7  | G8  | G9  | G10 |
|----|-----|-----|-----|-----|-----|-----|-----|-----|-----|-----|
| C1 | [M] | 2.1 | 0   | 4.5 | [M] | 7.3 | 8.9 | [M] | 3.4 | 2.5 |
| C2 | 1.1 | [M] | [M] | 3.4 | 2.3 | [M] | [M] | 0   | 2.9 | 0   |

Then, we filter both the [M] token and zero token for each sample. The resulting matrix is as:

|    | G1  | G2  | G3 | G4  | G5  | G6  | G7  | G8 | G9  | G10 |
|----|-----|-----|----|-----|-----|-----|-----|----|-----|-----|
| C1 |     | 2.1 |    | 4.5 |     | 7.3 | 8.9 |    | 3.4 | 2.5 |
| C2 | 1.1 |     |    | 3.4 | 2.3 |     |     |    | 2.9 |     |

Next, we concatenate the rest tokens sequentially and add the [PAD] tokens to match a max-length sample. In this step, some tokens only appear in one cell, introducing the gene inconsistency between the two samples. The generated matrix is as:

|    | Column1  | Column2  | Column3  | Column4  | Column5  | Column6  |
|----|----------|----------|----------|----------|----------|----------|
| C1 | 2.1(G2)  | 4.5(G4)  | 7.3(G6)  | 8.9(G7)  | 3.4(G9)  | 2.5(G10) |
| C2 | 1.1(G1)  | 3.4(G4)  | 2.3(G5)  | 2.9(G9)  | [PAD]    | [PAD]    |

# 3 Auto-discretization strategy evaluation

To validate the effectiveness of the expression value projection, we conducted an analysis of viewing the weight distribution pattern for continuous values. For each value, the auto-discretization block retrieves bucket embeddings (num=100) and combines them in a weighted manner to represent a targeting value. Different values correspond to a particular weight vector once finished training. Theoretically, close values have a similar weight vector distribution and distant values differ. The results showed that the normalized weight distribution of the close values exhibited smooth transitions and that of the distant values being clearly distinguishable (Figure 1). This supports the conclusion that the auto-discretization strategy effectively represents continuous values with high resolution while preserving relatively rich meaning.

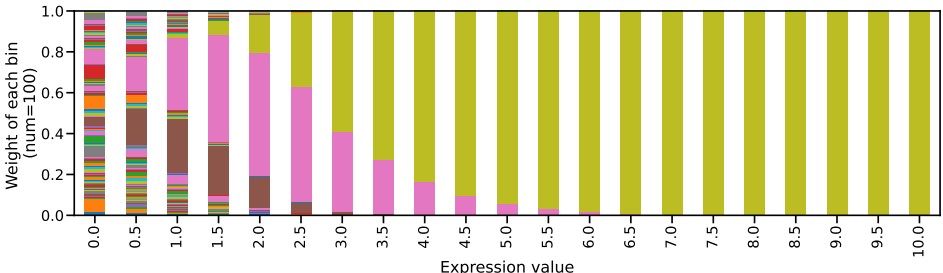

Figure 1: Weight distribution across bins for various expression values. The auto-discretization strategy was applied to each expression value in the range from 0 to 10, producing a corresponding weight vector with a length equal to the number of bins (100 in this case). The weight vectors were normalized to sum to 1 and visualized as stacked plots.

# 4 Clustering task evaluation and data sets

Cell clustering is an essential task for single-cell research, reflecting the ability of cell embeddings to remove noise and preserve biological signals. In the ablation experiments, we benchmarked the models' clustering performance on two cell-type annotated datasets.

**PBMC** This dataset is processed by Scanpy python library [7] and contains 2,638 cells. The cell types are annotated by the human with known markers and cover the major immune cells including B cells, CD4 &CD8 T cells, Monocytes, Dendritic cells, Megakaryocytes and NK cells.

**Experiments and Evaluation Metric.** For every single cell, the expression values are fed into the model and the max pooling layer is applied across all genes' output embedding to get a cell embedding. We then perform the usual single-cell clustering analysis step on these embeddings: 1) build the neighboring graph based on these embeddings and 2) use the Leiden algorithm to cluster the cell into groups. Since the Leiden algorithm requires resolution rather than the number of clusters, we used a dichotomy method to find an optimal resolution reaching the number of cell types given in the dataset.

Several evaluation metrics are applied to access the performance of the clustering results in different aspects, including Adjusted Rand index (ARI), Normalized Mutual Information (NMI), Homogeneity(HOMO), Completeness (CP), and Silhouette Coefficient (SIL). All these metrics are the higher the better. ARI and NMI measure the similarity of the clustering results from the statistics and information entropy theory view, respectively. HOMO and CP are intuitive metrics using conditional entropy analysis. HOMO measures how much the sample in a cluster are similar, and CP measures how much similar samples are put together. SIL measures the similarity of the embeddings to its cluster member compared to other clusters.

# 5   Acceleration strategy

The attention mechanism in masked language modeling is computationally expensive for long sequences, as time and space complexity grow quadratically along with sequence length. Though multiple attention architectures have been proposed to reduce the complexity to near linear, it's still slow to train large models with billions of data. We have adopted multiple techniques to boost the training speed as follows.

Since FP16 or BFLOAT16 Tensor Core has twice the computational throughput compared to TF32 on NVIDIA Ampere GPU, and, additionally, FP16 training also reduces residual memory consumption, xTrimoGene was conducted mainly with mixed-precision training strategy to optimize computational efficiency.

Distributed Data Parallelism is another training strategy used in our work, which handles large corpus on HPC clusters. In our setting, one single Ampere GPU provides sufficient amounts of memory for one model replica of billions of parameters performing forward and backward passes, and gradient accumulation is used to raise effective batch size to enhance large model training.

To scale up the model size, ZeRO-DP stage two [3] and checkpointing [1] techniques are experimentally tested in our setting. The results verified that both strategies reduce the model and residual state memory without expanding training time too much.

For the pre-trained xTrimoGene models, the memory consumption for inference with a sample of approximately 2000 non-zero expressed genes is approximately 50GB for the xTrimoGene-100M model and around 18GB for the xTrimoGene-10M model. It's worth noting that, in line with our pre-training settings, we conducted our tests using bf16 mode on an Nvidia A100 80G GPU. xTrimoGene-100M was trained on 5 million data points across 5 epochs using 64 A100 80G GPUs, with each epoch taking 12 hours.

# 6   Masking strategy

Table 1: Masking strategy for gene expression matrix. The gene expression matrix is masked by selecting a predetermined number of positions for prediction. $\sim 1{,}100$ positions, including $\sim600$ non-zero and $\sim540$ zero expressions, are masked in a matrix with $\sim20{,}000$ genes. The performance of the model is evaluated using Mean Squared Error (MSE) loss on these masked positions.

| Value | Masked | Unmasked | Total |
|---|---|---|---|
| $\neq 0$ | 600 | 1,400 | 2,000 |
| $= 0$ | 540 | 17,460 | 18,000 |
| sum | 1,140 (5.7%) | 18,860 (94.3%) | 20,000 (100%) |

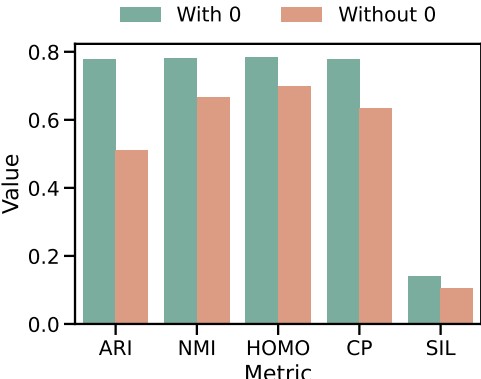

Figure 2: Cell clustering performance for xTrimoGene model considering masking zero (With 0) values or not (Without 0).

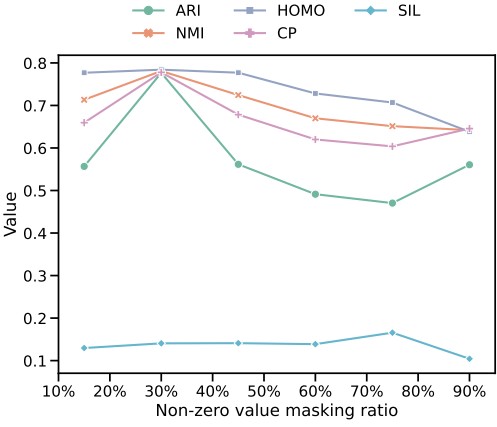

Figure 3: Model performance under different mask ratios of non-zero values. The cell clustering task is evaluated.

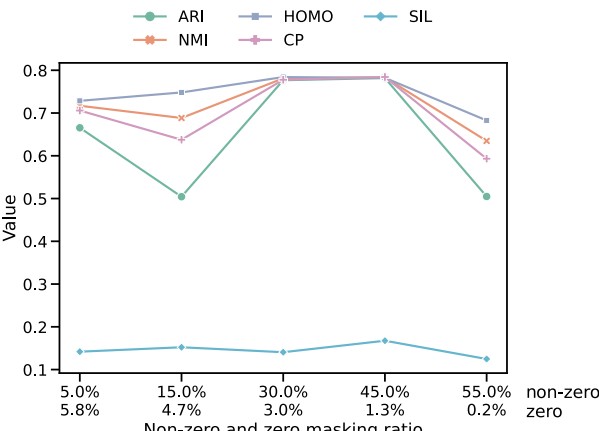

Figure 4: Cell clustering performance for xTrimoGene model with different masking zero, non-zero value ratio.

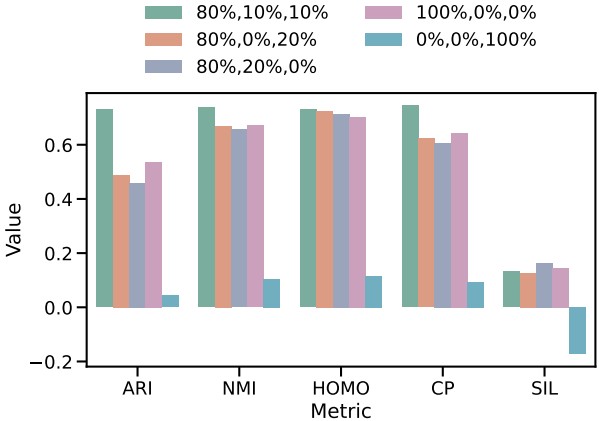

Figure 5: Comparison of performance for xTrimoGene model trained with different masking strategy. percentage1, percentage2, percentage3 denote corresponding replacing probability for three types of tokens: percentage1 for [MASK] token, percentage2 for random expression token and percentage for original token.

# 7 Scalability

Table 2: Size and hyper-parameters of the pre-trained models. All models are set to train on 5 million data sets and for 5 epochs.

| Model name | Parameter (M) | Encoder | | | Decoder | | |
|---|---|---|---|---|---|---|---|
| | | depth | heads | dim | depth | heads | dim |
| xTrimoGene-3M | 3 | 4 | 2 | 128 | 2 | 2 | 128 |
| xTrimoGene-10M | 10 | 4 | 8 | 256 | 2 | 4 | 256 |
| xTrimoGene-100M | 100 | 12 | 12 | 768 | 6 | 8 | 512 |

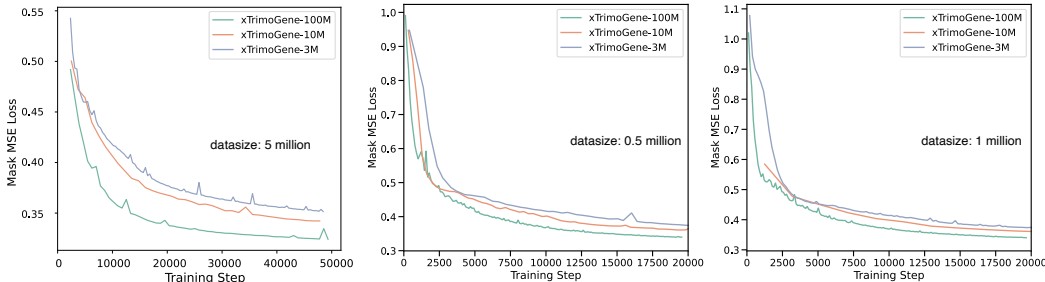

Figure 6: The learning curve of pre-trained xTrimoGene models with different parameter scale. The loss curve measures MSE for masked positions during the pre-training stage, and only the validation set is displayed.

# 8 Evaluation on downstream tasks

## 8.1 Cell type annotation task

We downloaded the Zheng68K expression matrix dataset from [8] and the Segerstolpe dataset from [5] and mapped the matrix to our reference gene list. Then, the dataset is split into training, validation and test sets with a ratio of 8:1:1. All the methods are trained on the training set and the best model is selected according to the performance on the validation set. Evaluation metrics (macro F1-score and Marco precision) are calculated for individual testing sets.

In the training process, the expression matrix is fed into the encoder of the xTrimoGene model and the gene embedding is obtained. Then we used a max-pooling layer to aggregate all gene embeddings into one cell embedding, and used a single linear layer to predict cell types from the embeddings.

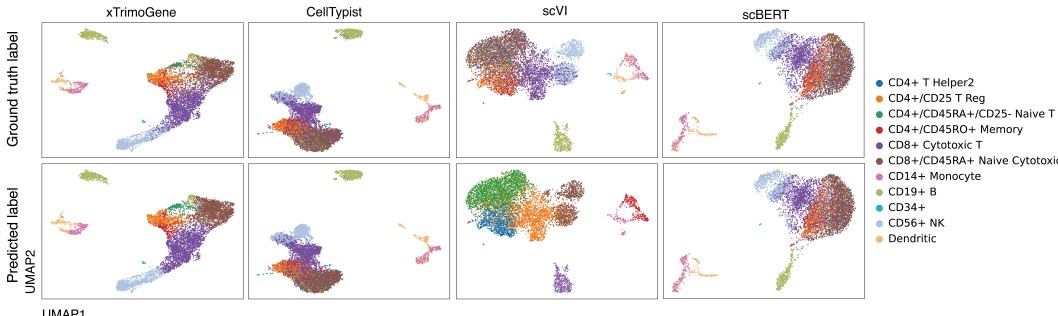

Figure 7: UMAP visualization of different models on the cell type annotation task (Zheng68K dataset). The dot in each panel denotes a cell that is colored by cell type. The two rows denote the ground truth and predicted cell type label, respectively.

## 8.2 Perturbation effect prediction

The Norman dataset is downloaded from a previous study [4]. The expression matrix data is mapped to our reference gene list. We reproduced the results of GEARS with original codes and settings (https://github.com/snap-stanford/GEARS). All the data processing is the same as GEARS, including data split, pre-post sample pairing strategy and evaluation metrics calculation.

While training GEARS with xTrimoGene, the expression matrix is fed into xTrimoGene and intermediate context embedding is obtained. The context embedding is then input to the co-expression graph network branch, all the other parts remain unchanged.

Table 3: The MSE of the top 20 deferentially expressed (DE) genes given by different pre-trained models on perturbation response prediction.

| Pre-trained model | Total | 1-gene | 2-gene(seen0) | 2-gene(seen1) | 2-gene(seen2) |
|---|---|---|---|---|---|
| xTrimoGene | 0.1983 | 0.1930 | 0.2385 | 0.2100 | 0.1286 |
| scBERT | 0.2231 | 0.2116 | 0.2581 | 0.2386 | 0.1522 |

## 8.3 Drug combination prediction

To test how xTrimoGene adapted to DeepDDS [6] for synergistic drug combination prediction, we first reproduced the DeepDDS algorithm. Both data and original codes are downloaded from Github repository (https://github.com/Sinwang404/DeepDDs/tree/master). We use data in "new_labels_0.csv" file for training and "independent_set" for testing. The genomic expression data are all mapped to our reference gene list. Models are trained 5 times and evaluated on the testing set. For all metrics, the averaged value and the standard deviation are reported. We keep the overall framework of DeepDDS

while testing xTrimoGene. The genomic expression matrix is fed to xTrimoGene and the intermediate context embedding is obtained. The embedding replaces raw expression profile for MLP branch input.

## 8.4 Rare cell detection

We also investigate how xTrimoGene behaves on these unseen data, we conduct the following evaluation analysis.

We first collected the scRNA-seq data from a previous study [2], which profiles human skin squamous cell carcinoma landscape with scRNA-seq and spatial transcriptomics technology simultaneously. The scRNA-seq data is not present during the xTrimoGene training process. Notably, the authors found that clustering the scRNA-seq data Figure (8, left) yields a novel cell subgroup named TSK (Tumor-Specific Keratinocyte), which is clearly in the tumor sample but not the normal sample.

We employed the tumor scRNA-seq data to explore whether xTrimoGene is robust to distinguish the cell subpopulations from others. The expression matrix is fed into xTrimoGene and the dumped context embedding is used for subsequent UMAP visualization. The results show that the TSK subgroup is clearly separated in Figure (8, right). More importantly, the two TSK subgroups are merged with xTrimoGene, demonstrating its generalization ability to generate good cell-specific embeddings for unseen data.

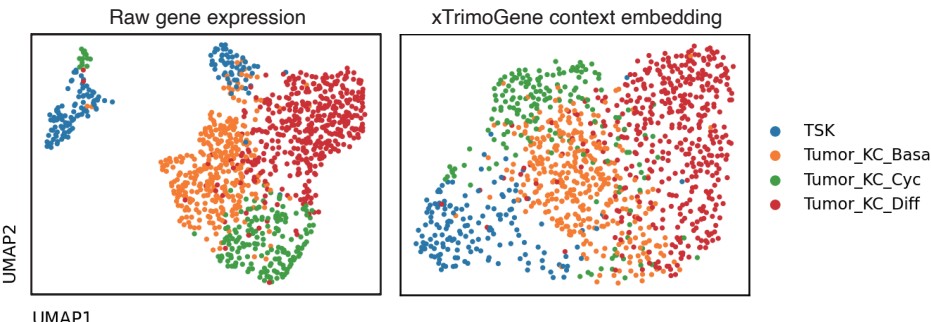

Figure 8: UMAP visualization of tumor scRNA-seq data with a novel TSK subpopulation [2]. The left panel denotes dimension reduction with raw normalized gene expression values, while the right panel with xTrimoGene dumped context embedding.

# 9  Website deployment of xTrimoGene model

xTrimoGene has been proven advantageous in gene representation and cell context embedding extraction. To facilitate its wide application for single-cell RNA-seq data analysis, we deployed the xTrimoGene model within the BioMap corporation. On the website, the xTrimoGene is implemented as a standard operator and serves multiple downstream tasks, including cell clustering, dimension reduction and batch removal Figure (9). The interactive page is user-friendly and feasible to evaluate performance with rich visualizations.

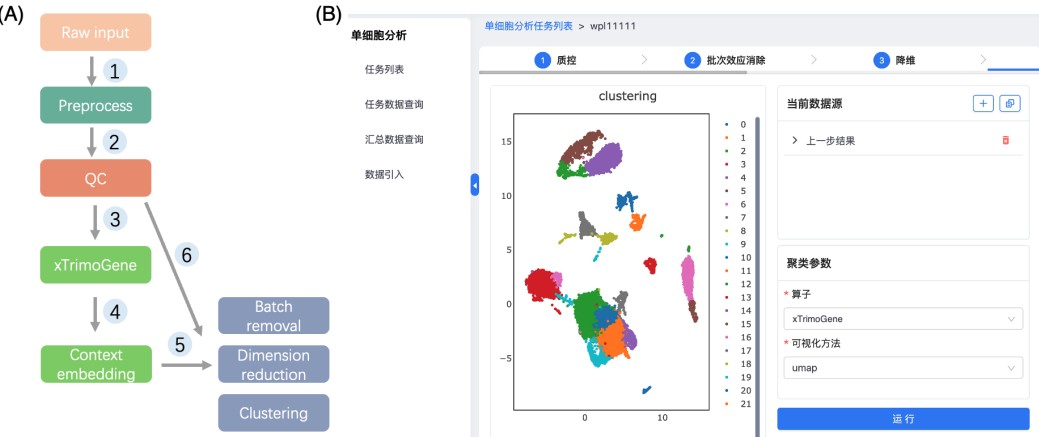

Figure 9: The deployment of xTrimoGene on a website is depicted in this figure. Figure A shows the overall pipeline, which includes the following steps: (1) User-uploaded raw input undergoes preprocessing and filtration through (2) quality control, (3) feeding the processed data into xTrimoGene for (4) context embedding extraction. The model supports multiple downstream applications such as (5) cell clustering, dimension reduction, and batch removal. The extracted expression profile can also be directly utilized by other algorithms. Figure B provides a snapshot of a clustering task in action using xTrimoGene's context embeddings.