# OpenReview forum: "xTrimoGene: An Efficient and Scalable Representation Learner for Single-Cell RNA-Seq Data"
_NeurIPS.cc/2023/Conference — NeurIPS 2023 poster_

### Official Review · Reviewer_N2nG · 2023-06-10

**Soundness:** 4 excellent
**Presentation:** 4 excellent
**Contribution:** 3 good
**Rating:** 7
**Confidence:** 4

**Summary:**

This paper proposes a scRNA-seq pretraining method, overcoming the scalability and resolution weaknesses of previous works. Experiments on various downstream tasks proves the efficiency and effectiveness of the proposed method.

**Strengths:**

The paper is well structured and easy to follow in general. The challenges it aims to tackle are also clear, namely, scalability and resolution. The proposed method is technically sound. Its performance and efficiency are evaluated on several downstream tasks. The experiments are thoroughly conducted to investigate the influence of different mask strategies and self-supervised objectives.

**Weaknesses:**

1. Would it be possible to further improve the efficiency by optimizing the padding operation? It seems that the padded zeros are also not useful in the encoding process.
2. The masking and reconstruction strategy, the auto-discretization module, and optimization objective seem to be a simple combination of previous works. This could slightly lower the novelty of the proposed method.
3. The embedding of the proposed method and existing works could be visualized to provided an intuitive understanding of the performance.

**Questions:**

Please refer to the weaknesses.

**Limitations:**

No significant limitations of the method are found.

---

> ### Author Rebuttal · Authors · 2023-08-09
>
> We thank the reviewer for the constructive comments. Here we showed additional analysis and discussions to further strengthen our work. If our response does not fully address your concerns, please post additional questions and we will be happy to have further discussions.
>
> **A1**: Introducing padding tokens within a batch is major compatible with parallel matrix operation. However, computes related to the [PAD] placeholder are useless as the corresponding attention is all masked out in implementation. In this scenario, diminishing computes on [PAD] token will further boost the training efficiency.
>
> One widely used strategy to avoid padding is sequence packing [1], which theoretically concatenates all the variable-length sequences into one sequence and then splits it into individual fragments with a fixed length. The resulting batches have a consistent shape and no further padding is needed. The method has been utilized in natural language modeling and protein sequence pre-training. However, the design is not applicable to xTrimoGene as gene expression pattern is cell specific, cross-concatenation will introduce noise signal and collapse the intrinsic cell representations.
>
> Another strategy called bucketing [2] tends to fit xTrimoGene more closely. While preparing the training set, one can sort the samples by expression pattern distribution. Specifically, the single-cell samples are sorted by dropout ratio and thus cells with similar non-zero length are divided into the same buckets. This will greatly reduce the needed [PAD] tokens within a batch and potentially accelerate the overall training process.
>
> [1] M Kosec et al, Packing: Towards 2x NLP BERT Acceleration. 2021
>
> [2] Tom Kocmi et al, Curriculum Learning and Minibatch Bucketing in Neural Machine Translation. 2017
>
> **A2**: Though individual component of the architecture has been explored before, the overall design is predominately motivated to align scRNA-seq data characteristics. Regarding the novelty, we would like to highlight the following contributions and advancements over previous methods.
>
> 1. xTrimoGene is the first asymmetrical encoder-decoder architecture to guide single-cell RNA-seq data pre-training. The scRNA-seq is highly sparse, thus using the encoder-only design tends to introduce huge amounts of redundant computations. The encoder in xTrimoGene concentrates on capturing intrinsic features from the most informative non-zero values, while the decoder additionally integrates zero-value genes to further tweak the gene-gene interactions. This strategy empowers the model to learn gene representations both efficiently and comprehensively.
>
> 2. Projection of expression value needs to maintain the continuous feature. Different from language tokens, the gene expression values in scRNA-seq data are continuous scalars, which typically indicate similar gene activity when they have similar values. To transform these scalars into high-dimensional tokens in the data matrix, a representation that preserves the continuous semantics is needed. We verified the effectiveness of the proposed auto-discretization strategy.
>
> 3. Training strategy setting is comprehensively ablated. We have conducted a series of ablation studies to validate the optimal training configurations. Concretely, we proved regression task (objective) is superior to the conventional classification one (Figure 2A). Instead of masking all positions randomly, we need to separately mask zero and non-zero values with a trade-off ratio (Appendix Figure 2,3,4).
>
> 4. We scaled the pre-trained model and achieved remarkable performances on multiple downstream tasks, including cell type annotation, perturbation prediction and synergistic drug combination prediction. The results demonstrate the generalization ability of xTrimoGene, which is expected to drive more advancements in other downstream task learning.
>
> In summary, xTrimoGene is developed to pre-train large-scale scRNA-seq data efficiently, where the underlying design is motivated and optimized to align scRNA-seq data characteristics. We envision the established framework is meaningful for further algorithm improvement.
>
> **A3**: For all the compared methods related to cell type annotation tasks, we investigated the embedding distributions with Zheng68K data set and uncovered a potential connection to model performance.
>
> Specifically, CellTypist and scVI are included for comparison versus xTrimoGene. We projected the embedding in a UMAP plot, where the cells are colored by either ground truth (upper row) or model predicted (bottom row) cell type labels (Figure R4). The model performance is correlated with the consistency level between the two plots.
>
> In contrast with CellTypist, xTrimoGene achieves a better performance in predicting the CD19+ B cell type (Figure R4, 1-1 v.s. 1-2). Meanwhile, the CellTypist tends to identify some CD8+ Cytotoxic T cells (2-1) as CD19+ B cell type (2-2). The CD8+ Cytotoxic T cell is the largest sub-population across all 11 cell types, potentially leading to the inaccurate assignment. scVI performs much worse than xTrimoGene, as a proportion of CD8+ Cytotoxic T cells (3-1) are detected as CD56+ NK cell types (3-2). It seems that two batches of Cytotoxic T cells are present and scVI is inferior to discriminate the batch effects. Notably, a smaller subgroup CD34+ cells (4-1) are nearly assigned as Dendritic (4-2) cell type incorrectly. This may suggest scVI has a limited resolution to separate rare cells.
>
> Visualization analysis of embedding is useful to interpret model behavior, especially when it remains challenging to establish explainability for deep models. The paradigm provides a convenient and intuitive manner to validate the performance and will be further utilized to decode the limitations and advantages of xTrimoGene.

---

> > ### Comment · Reviewer_N2nG · 2023-08-14
> > **Thanks for the responses**
> >
> > Thanks for the responses which solve my first two concerns. For the last concern, could the author also supply the UMAP comparison with scBERT?

---

> > > ### Author Response · Authors · 2023-08-16
> > > **Thank you for the feedback!**
> > >
> > > We will add the figure to our revised manuscript. But due to the constraints imposed by the discussion format, we are unable to revise or provide figures now. If you are now interested in the part and initiate a video request, we can format a figure video to the committee for access.
> > >
> > > Here is a summary of the scBERT UMAP: scBERT only identified 10 cell types across all 11 populations. And all the CD4+ T Helper2 cells are incorrectly assigned. The observation demonstrated scBERT performs inferior to xTrimoGene in capturing cell type specific embedding, especially for rare cell types.

---

### Official Review · Reviewer_TD4Y · 2023-06-12

**Soundness:** 3 good
**Presentation:** 3 good
**Contribution:** 3 good
**Rating:** 7
**Confidence:** 4

**Summary:**

In this study, the authors propose an asymmetric encoder-decoder transformer for scRNA-seq data, called xTrimoGene, for large scale dataset pre-training. Quantitative comparison with various SOTA approaches shows the advantage in efficiency and scalability. In addition, a series of downstream analysis further validate the performance of xTrimoGene.

**Strengths:**

1. The training process is accelerated by utilizing sparse input.

2. A novel auto-discretization strategy is introduced to map continuous expression values into a latent embedding space.

3. Considering its high accuracy and efficiency, the proposed large model xTrimoGene is a valuable contribution to the single-cell community.

4. The authors conducted validation of xTrimoGene using multiple downstream analysis tasks.

**Weaknesses:**

In table 2, compared with CellTypist, the propsoed method only obtained a marginal improvement. In this case, it will be better if the author can provide more analysis to show the advantage of xTrimoGene on this task. See questions for detailed suggestions.


**Questions:**

Compared with CellTypist, is there any biological insights can only be found by xTrimoGene from these datasets? Is there any specific circumstance in which xTrimoGene consistently demonstrates superior performance？

**Limitations:**

The limitations are not presented in the paper. Consider discussing the potential limitations or challenges associated with the proposed method. Acknowledging and addressing these factors would further enhance the study's robustness and reliability.

---

> ### Author Rebuttal · Authors · 2023-08-09
>
> We thank the reviewer for the constructive comments. Here we showed additional analysis and discussions to further strengthen our work, with a focus on the advantage over CellTypist. If our response does not fully address your concerns, please post additional questions and we will be happy to have further discussions.
>
> We have further explored the performance of xTrimoGene in comparison to CellTypist and demonstrated the following advantages and potential biological insights:
>
> 1. xTrimoGene is more robust to identify rare cell types than CellTypist.
>
> For the Zheng68K annotation task, xTrimoGene achieves a marginal improvement over CellTypist (F1-score: 0.7354 versus 0.7151). However, the boosting performance across individual cell type varies. Among all the profiled 11 different cell types, xTrimoGene gains a large margin (F1 score: 0.21 versus 0.0) for CD4+ T Helper2 cell type, which is the smallest subgroup in the total population. The observation suggests xTrimoGene is superior over CellTypist to detect and distinguish rare cell types.
>
> |Cell type|xTrimoGene-Precision | xTrimoGene-Recall | xTrimoGene-F1 | CellTypist-Precision | CellTypist-Recall | CellTypist-F1|
> | :--- | :---: | :---: | :---: | :---: | :---: | ---: |
> | CD14+ Monocyte (195) | 0.86 | 0.81 | 0.84 | 0.86 | 0.85 | 0.85 |
> | CD19+ B (558) | 0.96 | 0.81 | 0.88 | 0.90 | 0.84 | 0.87 |
> | CD34+ (19) | 0.90 | 0.95 | 0.92 | 1.00 | 0.84 | 0.91 |
> | CD4+ T Helper2 (9) | **0.20** | **0.22** | **0.21** | 0.00 | 0.00 | 0.00 |
> | CD4+/CD25 T Reg (612) | 0.78 | 0.68 | 0.73 | 0.72 | 0.69 | 0.71 |
> | CD4+/CD45RA+/CD25- Naive T (185) | 0.53 | 0.71 | 0.61 | 0.66 | 0.54 | 0.59 |
> | CD4+/CD45RO+ Memory (303) | 0.65 | 0.64 | 0.64 | 0.70 | 0.47 | 0.56 |
> | CD56+ NK (853) | 0.93 | 0.90 | 0.91 | 0.93 | 0.92 | 0.92 |
> | CD8+ Cytotoxic T (2031) | 0.87 | 0.88 | 0.87 | 0.86 | 0.83 | 0.84 |
> | CD8+/CD45RA+ Naive Cytotoxic (1636) | 0.84 | 0.91 | 0.88 | 0.80 | 0.94 | 0.87 |
> | Dendritic (194) | 0.80 | 0.84 | 0.82 | 0.84 | 0.83 | 0.83 |
>
>
>
>
> 2. xTrimoGene embedding reveals potential cell type specific gene-gene networks.
>
> xTrimoGene, as a pre-trained model, can generate unique embeddings which encapsulate intrinsic gene-gene relationships. In the Zheng68K dataset, after cell-type annotation, we conducted the differential gene expression analysis between B and non-B cell groups. And then we retrieved the top 10 genes specific to B cells and retrieved the context embedding for these 10 genes from xTrimoGene. These embeddings are used to construct a gene-gene network (Figure R3), where the edge represents gene embedding similarity. From the gene network, we found HLA genes family (HLA-) have higher similarities within each other, while gene CD74 has less similarity with others. The analysis illustrates that xTrimoGene embedding can be utilized to decipher cell type specific gene networks, which is not within the scope of CellTypist.
>
> In summary, xTrimoGene not only outshines in handling rare cell types but also offers the potential to decode gene-gene relationships, setting it apart from existing methods.

---

> > ### Comment · Reviewer_TD4Y · 2023-08-15
> >
> > Thanks for the response which mainly addresses my concerns. However, it will be better if the author can discuss some limitations of the proposed method.

---

> > > ### Author Response · Authors · 2023-08-16
> > > **Thank you for your comments!**
> > >
> > > Thank you for your feedback. Certain limitations exist for xTrimoGene and further work is desired to advance the design.
> > >
> > > 1. Not fully make use of cell meta information. In present, xTrimoGene major utilize gene expression values during the pre-training stage, overlooking varieties of other related meta information like sample condition (health/disease), cell type, tissue type, sequencing platform, etc. These rich annotations are biologically meaningful and highly correlated with the expression pattern within a cell. Incorporating such information is anticipated to learn intrinsic gene-gene regulations better and boost the model performance. However, more further experiments are needed to validate the hypothesis. An appropriate strategy to encode and integrate these discrete attributes may require extensive trials.
> > >
> > > 2. Engineering optimization. The memory consumption for inference with xTrimoGene-100M model is approximately 50GB, whose hardware requirement (Nvidia A100 80G GPU) is beyond some academic labs. Additionally, computes related to the [PAD] placeholder are useless as the corresponding attention is all masked out in implementation. In this scenario, computational or memory efficient engineering techniques tend to advance the model pre-training and application.
> > >
> > > Thank you again for pointing this out and making our work more comprehensive. We will include the discussions in our future version.

---

### Official Review · Reviewer_MmMn · 2023-07-04

**Soundness:** 3 good
**Presentation:** 3 good
**Contribution:** 3 good
**Rating:** 5
**Confidence:** 4

**Summary:**

The manuscript proposes an adaptation of the key contributions of [1] to the scRNA-seq data modality, which is dubbed xTrimoGene. Additionally, the authors introduce a new embedding encoder, their "auto-discretization strategy". Several ablation experiments motivate specific choices in the model. Finally, xTrimoGene is tested on three downstream tasks.

**Strengths:**

The model is described (for the most part) clearly, so that the paper is easy to follow. Utilizing the sparsity of scRNA-seq data is important in the context of transformers and, therefore, adapting the ideas from [1] to this modality seems like a very good match. I also appreciated the ablation experiments because they justify specific choices for the model and make the develoment process more transparent.

**Weaknesses:**


**Connection to [1]:**
The paper is for the most part a (useful) adaptation of the ideas of [1] to the scRNA-seq domain. This is mentioned in one sentence on page 5 (l 156), which, in my opinion, under emphasizes the connection. I appreciate that additional steps, such as determining the correct size of the mask and reducing the frequency of masking zeros are necessary to transfer the ideas to scRNA-seq data. Nevertheless, I strongly suggest putting references to [1] more prominently and earlier in the paper, e.g., into the abstract, the introduction and the beginning of Section 3 (at least beginning of section 3.1).


**Exposition of the auto-discretization strategy:**
While the paper is overall clearly written, I did not understand several aspects about the auto-discretization strategy.
First, why is discretization necessary in the first place; unnormalized count data would already be discrete, so why not use this? Also, why is discreteness necessary?
Second, does the proposed method actually discretize? According to ll168 ff it is just a weighted sum of vectors and Appendix Figure 1 confirms this empirically at least for low expression values.
Third, I did not get the description in the first paragraph of 3.3. My main question is what the shapes of the various w's and v's are in that paragraph. My best guess is that $v_1, v_2, v_3, v_4 \in \mathbb{R}^{c\times b}$. But then the product $EXP_{lookup} \cdot v_4$ is the wrong way around and would only produce output in $\mathbb{R}^{c\times d}$, while from Fig 1 I understand that the output should be in $\mathbb{R}^{c \times m \times d}$, which makes sense as m is the sequence length, crucial for the transformer. Also, I wonder why the two products of $v_2$ are not collapsed into a single one in line 167. Should perhaps one of the two $v_2$'s be a $v_1$?
When the authors write in ll 168 ff "The final output is a weighted combination of individual embeddings from the look-up table [...], where the weights are learnable parameters.", I wonder whether the entries of the look-up table are themselves learnable, or just the weights $w_1, w_2$. Also, why is $EXP_{lookup}$ called a look-up table, when actually a weighted sum of its rows is returned? I assume that the softmax performs a soft version of discritization, with a winner-takes-(almost-)all effect. But at least for low expression values the output really is not discrete.

**Ablation of the relative masking frequency of zeros:**
I agree that zeros need to be masked much less frequently than non-zero entries of the gene expression matrix. The authors' choice of masking an "almost equal number of positions for zero and non-zero positions" is plausible (Why "almost" here? What is the precise masking strategy?). I would be curious as to how the downstream results change for other ratios. I.e. I suggest a plot like in the Fig 3 of the appendix, in which the total masking ratio is kept at 30% but the frequency of masking zeros and non-zeros is changed. Perhaps something similar is depicted in Fig 4 of the appendix, which I did not understand (Could you please explain the three percentages in more detail?).

**Order of subsections in section 5:**
The main experimental results appear in subsections 5.4, which also describes the downstream tasks for the first time and compares to competitors. Sections 5.1-5.3 are valuable in that they analyze the model in more depth and perform ablations. Some ablations are even based on the task described only later in 5.4. Therefore, I would suggest to move the most important subsection, 5.4., to the beginning of section 5. In addition, one might also put the experiments of Fig 2 into section 5 (after the downstream experiments have been described) because 2A also relies on the downstream task. This way, the reader would be familiar with the downstream tasks before they are used to compare models.

**Comparison to other models:**
For the cell type annotation task many methods were used to produce a representation which was subsequently used for clustering. I guess that nearly all of these methods (for sure scBERT, scVI) can also be used in conjunction with GEARS and DeepDDS in much that same way as xTrimoGene is used for perturbation prediction and drug prediction, respectively. While the current experiments for these two task show that xTrimoGene representations offer benefits compared to raw gene expressions, it would strengthen the paper if xTrimoGene also outperformed, say, a combination of scBERT representations and GEARS.



**Minor:**
- In line 15 of the abstract the downstream task is described as "cell classification". This sounds as if it was supervised. But according to the appendix unsupervised Leiden clustering is used. Please change this to "cell type annotation" or "cell clustering".
- In line 19 the term AI4Science is introduced but never used. Perhaps it can be omitted?
- In l 84 the authors say that gene expression values in scRNA-seq data are continuous scalars. But the raw data is counts, i.e., discrete. This confused me when reading the paper the first time. From l 103 onwards, they speak of *normalized* gene expressions, which indeed are real numbers and not necessarily natural numbers. Please add "normalized" or "pre-processed" to l 84. The pre-processing is discussed in the appendix, but not linked from the paper. Perhaps it would be useful to include a cross-reference at the beginning of section 3, as a first step of the pipeline.
- Line 131 refers to the auto-discretization strategy "discussed previously", but the auto-discretization strategy is only discussed later in section 3.3.
- It would be useful to mention the key aspects of the large pre-training dataset at the beginning of section 5 (size, number of genes, that it was scraped from GEO) at the beginning of section 5.
- It would be great if Table 1 was extended by the runtime and memory consumption of each model.
- Line 271 and Table 2 speak of Zheng68K, which has 68K cells, while appendix l 43 claims the PBMC dataset had only ~2k cells.
- Line 162: should it be $V\in \mathbb{R}^{c\times m}$ rather than $V\in\mathbb{R}^{c\times n}$?
- Fig 3 panel A's caption and main caption "Sparse level" --> "Sparsity level"
- First line of Fig 2: Typo in "Performance" and missing hyphen in "auto-discretization".
- Line 227: word order "other two" --> "two other"
- Line 228 "comparison, three models" --> "comparison, all three models"

**Reference:**
[1] He, Kaiming, et al. "Masked autoencoders are scalable vision learners." Proceedings of the IEEE/CVF Conference on Computer Vision and Pattern Recognition. 2022.




**Questions:**

- How is the name "xTrimoGene" motivated? Please include an explanation of this non-obvious name.
- What are the numbers in brackets in line 154?
- What is the range of the sum in Eq 5? Just the masked entries of the gene experession matrix (similar to [1]) or all entries? The normalization prefactor does not match either.
- How is classification in the classification task mode of Fig 2B) performed? I understand that the output is continuous and not discrete, so that I do not see what the classes would be. For this reason, I have accepted the use of MSE as loss immediately. Nevertheless, performing the ablation is appreciated.
- Why was the comparison in Table 2 only performed for the 10M parameter model and not the 100M parameter model? The latter has lower validation loss, so it should perform even better, right?
- The model was trained on a very large dataset of human gene expressions. Is there any hope for it to transfer to other species?
- I understood that the gene embedding was akin to positional embeddings in NLP. Therefore, would have assumed that it only changes with the gene, not the cell. Is this correct? If so, why are the rows in Fig 1 upper right corner not all the same?
- Are $I_{masked}$ and $I_{zero}$ in eq (3) also the sums of the auto-discretization encoder and the gene encoder? Or are the pre-processed gene expression levels directly used here?

**Limitations:**

Limitations are not explicitly discussed. Code is not part of the submission, but is promised to be released on GitHub alongside the pre-trained model, which is particularly interesting.

---

> ### Author Rebuttal · Authors · 2023-08-10
>
> **WA1**: We are grateful for all your comments and will revise our manuscript to make more discussion and emphasize the connection. In our scenario, this asymmetric encoder-decoder architecture is not only efficient but also designed for handling the high sparsity in scRNA-seq data. Specifically, we feed the non-zero expressed genes into the encoder and let the zero expressed genes only be processed by the decoder.
>
> **WA2**：
> 1. Using the count values to query embeddings can cause the model to overly capture noise since the MSE loss is sensitive to the data scale. This noise makes it difficult for model optimization. Therefore, the normalized data is more applicable for model training and a "coarser-grained" representation strategy- discretization or quantization, is needed to make the continuous values be mapped into similar embeddings.
> 2. Our method is more aptly described as a form of soft discretization. This contrasts with traditional methods of hard discretization, which can lead to problems like Similar value But Dissimilar embedding (SBD) as highlighted in ll53-ll55. The soft discretization, allows for flexibility, effectively addressing the SBD challenge. Additionally, our Auto-discretization method is differentiable, supporting end-to-end training. Also, the embedding methods for numerous or continuous values can be observed in other fields[1,2]. Appendix Figure 1 shows bin distribution for low-expression values. This is because the majority of gene expressions in our training data fall within the 1-2 range. Those exceeding a value of 4 are relatively rare. We are sorry for the confusion and the shapes for all the matrices are added in the paragraph.[1] AutoEmb: Automated ... Recommendations
> [2] AutoDis: An Embedding ...  Prediction
>
> 3. Both the look-up table entries and the weights, w1 and w2, are learnable. The process can be viewed as a soft discretization, and while the outcome is a weighted sum of the embedding table (refer to Q2), we label it as a "look-up" due to its meaning of mapping raw data to its respective embedding, we are sorry for this confusion. The low expression values are not strictly discrete since the data distribution is not uniform.
>
> **WA3**:
> In total, 1,140 (5.7\% of all genes) entries are masked, which on average includes 600 non-zero entries (30\% of total non-zero value genes) and 540 zero entries (3\% of total zero value genes).
>
> We ablated the ratio of the zero under the same total masking ratios/numbers (5.7\% of all genes, n=1,140) as in the public comments experiment no.4. For each setting, we trained a 10M parameter model over 5 million data for 50,000 steps. Then, we evaluated the model on PBMC3K downstream cell clustering task. As the zeros masking ratio increases, the performance tends to be better and then achieves a plateau (1.3\% - 3.0\%). The current zeros masking ratio lies in the plateau interval.
>
> For appendix Figure 4, we aim to investigate if all the masked values should be replaced with a [MASK] token. The three percentages (80\%,10\%,10\%) represent the probability of masked values being replaced with a [MASK] token, a random expression token, and the original token, respectively.
>
> **WA4**:
> Thank you for the proposed logic to organize the work, we have revised the main text accordingly.
>
> **WA5**:
> Yes, the comparison can also be applied to other tasks with produced cell embedding. Here we investigate whether xTrimoGene is advantageous over scBERT on the perturbation effect prediction task. Similar to xTrimoGene, the generated embedding from scBERT is in conjunction with GEARS. As shown in the public comments experiment no.5, xTrimoGene achieves a lower MSE value than scBERT across all metrics, demonstrating its superior efficiency in generating cellular context embedding under different biological conditions.
>
> **Minor**: For minor comments, we revised the manuscript accordingly. And for datasets, we used two datasets named PBMC and Zheng68K with ~2k and 68k cells, respectively.
>
> **Q2**: The numbers in brackets are the absolute depth/head value between the encoder and decoder. We deleted the numbers and added the reference to Appendix Table 2 for clarification.
>
> **Q3**: Equation 5 defines the loss at masked positions (including zero and non-zero entries), whose range and prefactor vary across different samples.
>
> **Q4**: The ablation studies across the paper are all conducted with model context embedding. Concretely, we fed the PBMC3K expression matrix into the evaluated model and dumped the context embedding. Then, we utilized the embedding to cluster all the cells and calculated the clustering metrics. In the scenario of classification pre-training mode (Figure 2B), the expression value is rounded into an integer (each integer as a token) and the pre-training objective is to predict the token (which is discrete). After the pre-training stage is finished, the model is evaluated on the ablation cell clustering task as aforementioned with the exception that the PBMC3K expression matrix are rounded integers.
>
> **Q5**: Sorry for the typo, the result is from xTrimoGene-100M model.
>
> **Q6**:  We can fine-tune the pre-trained model in mice as the scRNA data is comparable with humans. Upon fine-tuning over a large amount of data, the model is possible to capture the intrinsic species-specific regulations and learn good representations for species-specific genes.
>
> **Q7**:  Yes, the gene embeddings are changed along with genes instead of the cell. For Fig 1 upper right corner part, we are retrieving the expression and gene embeddings for the unmasked-only matrix. The matrix has been processed with filtering and padding steps. In the former step, we filtered out the zeros and masked entries, which positions/orders are not consistent across different cells. Thus, we utilized a color gradient to denote the gene differences.
>
> **Q8**: The former, they are also converted to embedding similar to unmasked entries, instead of pre-processed value.

---

> > ### Comment · Reviewer_MmMn · 2023-08-11
> >
> > Thank you for the detailed response!
> >
> > **WA2:**
> > 1. Thanks for explaining.
> > 2. Unfortunately, I am no expert on these soft or hard discretization methods. I would appreciate a more basic explanation of this step. Why is any form of expression embedding computed at all? Why not just multiply the embeddings of genes expressed in a cell by their (normalized) expression in that cell? Is the number of tokens equal to the number of bins? If not, how do the two concepts relate? Does a token still correspond to a unique gene, or to a mixture of genes? The reason for only using 100 tokens is to make the computations tractable (because the attention matrices will be of shape $(100, 100)$ rather than (20k, 20k)), right?
> > 3. Thanks for clarifying.
> >
> > **WA3:**
> > Thanks for reporting this sensitivity analysis on the masking ratio! I think it justifies your choice and thus strengthens the paper.
> >
> > I understand App. Fig. 4 better now, thanks. Which of the five settings explored in this figure do you use in your main experiments? What is the rational behind replacing a masked token by the original token? This amounts to just not masking it, right?
> >
> >
> > **WA5:**
> > Excellent, thanks for checking this!
> >
> >
> > **Q3:**
> > I see. Perhaps writing something like
> >
> > >$\sum_{i=1}^c \sum_{j\in\mathcal{M}_i}$, where $\mathcal{M}_i$ is the set of $n-m$ masked entries in cell $i$
> >
> >  would help. Why does the size of the mask differ between samples? I thought you always masked 1140 genes per cell?
> >
> > **Q4:**
> > Thanks for explaining the classification pretraining mode in more detail. Indeed, normal masked language learning is phrased as a classification task over the type of token (i.e., which word is it?). My intuition was that the scRNA-seq analogue to word is gene and that the frequency of a word is akin to the expression value of a gene. Therefore, I am surprised that you predict a discretized expression value rather than the gene type. I would have rather expected a comparison in which the gene identity is predicted. But I do not think this is an important issue.
> >
> >
> > **Q7:**
> > Ok, but if the gene embeddings only change by gene not by cell, then I still do not understand why the columns of the right stack of matrices in Fig 1 top right corner are not constant. In Fig 1 a) of the scBERT paper the rows of the gene expression matrix are constant (it is rows instead of columns because they work with the transposed setup).

---

> > > ### Author Response · Authors · 2023-08-14
> > > **Thank you for your comments!**
> > >
> > > Thank you for your valuable feedback. Here we provide more explanations and examples to clarify the auto-discretization and model design. If our response does not fully address your concerns, please post additional questions and we will be happy to have further discussions.
> > >
> > > **WA2**: Simply multiplying gene embeddings by their normalized expression might seem straightforward, but it presents certain challenges.
> > >
> > > Due to large amounts of zero expression values in the data, the multiplication would result in numerous zero-valued embeddings, rendering such embeddings uninformative.
> > > Whereas in the xTrimoGene encoder which exclusively processes non-zero values, expression value multiplication may be applicable. However, it still has the risk of missing key regulatory relationships. Some genes (such as some transcription factors) can significantly influence the expression dynamics of other genes, even if their own expression is not high. The multiplication will reduce the importance level of these low-expressed genes, yielding biased and incomplete regulatory relationships.
> > >
> > > No, the number of tokens is not equal to the number of bins. A token is the input unit of the transformer, and the number of tokens for each sample is 19,264. Each token is the sum of a gene embedding and a value embedding (both d dimension).  Value embedding is obtained from the auto-discretization module and is weighted summarization of 100 embeddings (bins) from a look-up table $EXP_{loopup} \in \mathbb{R}^{d \times b}$, where $b=100$ indicate the number of embeddings in the table and refers to the bin number. So the attention matrix is of shape (number of non-zero \& non-mask genes, number of non-zero \& non-mask genes) in the encoder and (19,264, 19,264) in the decoder.
> > >
> > > **WA3**: Based on the App. Fig 4, we opted for the first configuration (80\%,10\%,10\%).
> > >
> > > The design is to minimize the disparity between the pre-training phase and the fine-tuning downstream tasks. Given that the inputs during downstream fine-tuning are not subjected to masking, it becomes pivotal to ensure that the pre-training objective encompasses positions that do not solely consist of [MASK] tokens. The strategy has been proven effective in natural language pre-training [1], which results are consistent with our ablation study.
> > >
> > > [1] Devlin et al. BERT.... 2019
> > >
> > > **Q3**: Thank you for the suggestion. We will revise this.
> > > 1,140 is the approximate average masked number of the total samples and is not fixed. Actually, in the implementation, we fixed the mask ratio (zero: 30\%, non-zero: 3\%). So the mask size $m$ can be defined as:
> > >
> > > $$m=a\times 0.3 + b\times 0.03$$
> > >
> > > where $a$ and $b$ is the number of zero and non-zero value, respectively. These two values are different across samples, leading to a non-constant absolute mask size $m$.
> > >
> > > **Q4**:The gene can be akin to a word and the expression value to the word frequency is indeed the case.
> > > By following this intuition, one possible way is to repeat each gene multiple times, i.e., extending times by expression values. However, two factors may limit the extending alignment: (1) Though the frequency of the gene is clear, the exact position of the added gene is not known. For instance, extended genes (e.g., G1,G2,G3 with expression value 2,1,2) can be sequentially (G1G1G2G3G3) or cycling concatenated (G1G2G3G1G3), which represents two distinct gene sentences and impact the modeling. (2) If each gene is extended multiple times (maybe several to tens to hundreds), the resulting sentence will achieve a large length, constraining the pre-training efficiency.
> > >
> > > **Q7**: We take the below example to demonstrate the processing flow.
> > >
> > > Assume an expression value matrix with 2 cells and 10 genes. First, we masked a portion of values and the generated matrix is as:
> > > ||G1|G2|G3|G4|G5|G6|G7|G8|G9|G10|
> > > |-|-|-|-|-|-|-|-|-|-|-|
> > > |C1|M|2.1|0|4.5|M|7.3|8.9|M|3.4|2.5|
> > > |C2|1.1|M|M|3.4|2.3|M|M|0|2.9|0|
> > >
> > > Then, we filter both the M token and zero token for each sample, concatenate the rest tokens sequentially and add the PAD tokens to match max-length sample. The generated matrix (unmasked-only matrix in Fig1) is as:
> > > ||Column1|Column2|Column3|Column4|Column5|Column6|
> > > |-|-|-|-|-|-|-|
> > > |C1|2.1(G2)|4.5(G4)|7.3(G6)|8.9(G7)|3.4(G9)|2.5(G10)|
> > > |C2|1.1(G1)|3.4(G4)|2.3(G5)|2.9(G9)|PAD|PAD|
> > >
> > > The generated matrix is utilized to retrieve both expression and gene embeddings. The genes in Column1 are different (G2 versus G1) across the two samples, while genes in Column2 are the same (G4).
> > >
> > > Conversely, the gene rows are constant in scBERT. The discrepancies in these schemes primarily arise from differences in architectural design. scBERT operates as an encoder-only framework, where all genes are involved in computation across each Transformer layer, thereby preserving the same gene order.

---

> > > > ### Comment · Reviewer_MmMn · 2023-08-15
> > > >
> > > > **WA2:**
> > > > Thank you for the detailed answer! Your explanation as to why a dedicated value embedder is necessary was helpful as was the discussion on the various dimensions. In particular, you mentioned that $b$ is the number of bins, which is different from the number of tokens. Please update l164, in which it says
> > > >
> > > > >  $b$ is the number of tokens (default 100)
> > > >
> > > > I only have some small remaining questions. Why was the module dubbed (soft) "discretization". My current understanding is that it simply is an embedding computed from the normalized expression vector of each cell. There is no discretization, right? The second question is more to check my understanding: The value embedding is the same for all (#non-zero genes + #non-masked) tokens of a cell and only depends on the full (normalized) expression vector of that cell, but not on the expressions of other cells, right?
> > > >
> > > >
> > > > **WA3:**
> > > > I did not realize that this 80%/10%/10% strategy was already used in the original BERT model. This provides a good grounding of your choice in the literature.
> > > >
> > > > **Q3:**
> > > > Thanks for clarifying and for revising the equation.
> > > >
> > > > **Q4:**
> > > > I think there was misunderstanding. My question was not about whether / how to represent a gene multiple times. Rather it was about whether the expression level or the gene identity is predicted in the classification setting. I understood your explanation as saying that you try to predict the (rounded) expression value. In my mind, when translated to language modelling this would amount to predicting how frequent a word was (i.e. which non-zero integer expression is predicted). That is why in my previous comment, I was surprised that you motivate your classification setting by the classification setting in masked language modelling, where one predicts which word it is, not how often that word appeared. I appreciate that language modelling and your masked training for scRNA-seq are not entirely comparable, since when you mask a gene, you mask *all* of its expressions, while in language modelling each masked position masks only a single occurrence of a word, not all occurrences of that word in the sequence. Nevertheless, I understand your classification setting now. Thank you for elaborating!
> > > >
> > > >
> > > > **Q7:**
> > > > Thank you very much for detailing the processing flow! I understand what happens in your method and the difference to scBERT much better now. Perhaps such an example, e.g., in the appendix might be useful to other readers as well.

---

> > > > > ### Author Response · Authors · 2023-08-16
> > > > > **Thank you for your comments!**
> > > > >
> > > > > Thank you for your valuable feedback.
> > > > >
> > > > > **WA2**:
> > > > > We are sorry for the misleading in l164, the word "tokens" will be modified to "bins".  For the two questions:
> > > > > 1. Yes, the expression value is projected into an embedding directly and no step to split the values into contiguous (discrete) intervals.
> > > > > 2. The value embedding is consistent between genes or cells, i.e., if two genes (from a cell or two cells) have the same expression value then their value embedding is the same. Yes, the embedding depends on individual cell as the attention is calculated within the cell, regardless of the other cells.
> > > > >
> > > > > **Q7**: We will incorporate the flow schemes into the appendix.
> > > > >
> > > > > Thank you again for your constructive feedback, helping us to clarify all the designs and concepts more clearly.

---

> > > > > > ### Comment · Reviewer_MmMn · 2023-08-21
> > > > > >
> > > > > > Thank you for the detailed discussion, which clarified important aspects. I recommend to improve the description of the auto-discretization scheme more accessible, along the lines of our discussion. I maintain my score.

---

### Official Review · Reviewer_9yec · 2023-07-06

**Soundness:** 3 good
**Presentation:** 3 good
**Contribution:** 3 good
**Rating:** 7
**Confidence:** 4

**Summary:**

Here the authors propose xTrimoGene, a scalable transformer-based model for learning representations of scRNA-seq data. The authors demonstrate that their proposed method is more computationally efficient than alternatives for training transformers on scRNA-seq data, and they also validate their model on cell type annotation and perturbation response prediction tasks.

**Strengths:**

* **Clarity**: I found the manuscript very well organized and the writing easy to follow. Well done!
* **Significance**: There has been much recent excitement about the potential utility of applying transformer architectures to scRNA-seq data. Here the authors demonstrate a method for making the training of such models more efficient, and also they demonstrate that the embeddings from such models are useful for multiple downstream tasks. Thus, I believe this work is significant.
* **Originality**: The authors' proposed strategy for training xTrimoGene is, to my knowledge, novel (though I note that I am not an expert in transformer training strategies).

**Weaknesses:**

The results presented by the authors are promising, and I indeed enjoyed reading the paper. However, I do have some minor issues that I would like to see clarified during the rebuttal period before I can give a recommendation of acceptance. If the authors are able to address my concerns, I would be happy to raise my score:

* **Was data leakage avoided?**: The authors briefly describe their data collection pipeline in Section 1 of the Appendix. However, this description is a bit sparse (just mentioning that data was collected from the GEO). Were any precautions taken to avoid data leakage issues in the evaluation of xTrimoGene on downstream tasks? That is, did the authors ensure that any test cells from train/test splits e.g. on the perturbation prediction task were not present in the data used to pretrain xTrimoGene?
* **Proper modeling of scRNA-seq count distributions**: Previous works (e.g. the Deep Count Autoencoder https://www.nature.com/articles/s41467-018-07931-2) have found significantly better preservation of biological variation in latent representations of scRNA-seq data by directly modeling scRNA-seq counts (e.g. using a negative binomial or zero-inflated negative binomial distribution) compared to minimizing MSE loss for normalized counts. However, xTrimoGene opts for the latter. I think it would be valuable to know how this choice affects the model's latent representation. An ablation study comparing modeling the raw counts versus normalized counts would be very interesting here. However, if the authors are unable to perform such an experiment in the rebuttal period, performing an (easier) experiment similar to that presented in Figure 2 of the Deep Count Autoencoder paper (depicting how increasing levels of noise can potentially result in low-quality latent representations) would also provide valuable results.
* **How to handle batch effects?**: A classic problem in scRNA-seq analysis is integrating datasets from multiple batches that may have systematic differences unrelated to any underlying biology. Given that xTrimoGene's pretraining procedure does not account for batch effects, could the authors discuss how would one analyze data from different batches with xTrimoGene?
* **Rare cell types**: A potential issue with pretrained models like xTrimoGene is that they may misbehave when applied to new datasets with phenomena not present in the training set (e.g. new cell types or tissues not seen in the training data). Did the authors explore xTrimoGene's behavior in such a scenario? A new experiment investigating this behavior would be useful.
* **Memory/hardware requirements?**: Could the authors provide additional details on the hardware/memory requirements for loading and performing inference with a pretrained xTrimoGene model?

**Questions:**

See "Weaknesses" section.

**Limitations:**

I would like to see the authors clarify some potential limitations with their method (see "Weaknesses" for specific questions). I do not forsee any negative societal impacts resulting from this work.

---

> ### Author Rebuttal · Authors · 2023-08-09
>
> We thank the reviewer for the constructive comments. We have shown additional discussions and experiments to further strengthen our work. The point-by-point responses to the comments are as follows. If our response does not fully address your concerns, please post additional questions and we will be happy to have further discussions.
>
> > **Q1: Was data leakage avoided?**
>
> **A1**: We did not specifically introduce a precaution strategy during the pre-training data preparation.
>
> However, we suppose this has no significant influence on the downstream task evaluation for the following two reasons:
>
> 1. Nature of Training:
>    - The pre-training process is self-supervised, focused solely on gene expression values, while other meta-information like cell type, tissue type, sequencing platform, etc., being disregarded.
>    - Conversely, the downstream tasks (such as perturbation prediction) are supervised, employing distinct cell label information or relationship of pre-post perturbed cells.
>
> 2. Learning Focus:
>    - During pre-training, the model is primed to understand intrinsic gene relationships without any exposure to label information. While for the downstream task, the model learns the relationship between gene expressions and cellular labels.
>
> Hence, even if there were overlaps between pre-trained and downstream evaluation data, the lack of exposure to label information during pre-training should safeguard against any significant data leakage effects.
>
> > **Q2: Proper modeling of scRNA-seq count distributions.**
>
> **A2**: The modeling of xTrimoGene is different from DCA. DCA uses the observed gene expression to predict the parameters in the ZINB distribution. In contrast, xTrimoGene is a masked autoencoder [1], i.e., recovering the unobserved gene expression values depends on other genes within a cell. MSE loss is generally a default choice. However, the pre-training data distribution of raw counts is very skewed, where most values are small but a few are very large, using MSE may cause the model to focus too much on those large values. Therefore, we use the normalized expression value.
>
> Following the suggestion, we explored an ablation study to compare the raw counts versus normalized expression values in our modeling. Concretely, we trained a 10M parameter model with raw counts as input, while all the other configurations keep consistent as the xTrimoGene-10M model. Then, the pre-trained model is evaluated on the PBMC3K clustering task, in which the raw count matrix is fed into the model to obtain the context embedding for subsequent analysis. The results show that pre-training with normalized gene expression value achieves a better performance than raw counts, which is consistent with our aforementioned assumption.
>
> |Input| ARI| NMI| HOMO| CP|SIL|
> |-|-|-|-|-|-|
> | Normalized value | 0.7767|0.7810|0.7841|0.7778 | 0.1406 |
> | Raw count| 0.6575| 0.7318|0.7459| 0.7183|0.1086|
>
> [1] He et al. Masked Autoencoders Are Scalable Vision Learners. 2021
>
> > **Q3: How to handle batch effects?**
>
> **A3**: For xTrimoGene, two strategies could handle batch effects:
>
> 1. Fine-Tuning with Batch Information: xTrimoGene can be fine-tuned with user-specific datasets incorporating batch information. We achieve this by:
> (1) Converting the batch ID to embeddings via a lookup table.
> (2) Summing these batch embeddings with the original value and gene embeddings. This aggregated input is then processed by the transformers. During fine-tuning, gene expression embeddings from different datasets are enriched with their corresponding batch embeddings.
> (3) At the inference stage, to harmonize cells from different batches, a consistent batch embedding can be utilized, ensuring the output remains consistent across batches.
>
> 2. Integration with Other Batch Correction Techniques: One could dump embeddings from xTrimoGene and feed them into other methods.
> For this strategy, we used the pancreas dataset (comprising 8 batches and is a benchmark for batch integration) as an example. We subsampled this dataset to 3k cells for efficiency.
> After getting embeddings from xTrimoGene, we employed a lightweight method, BBKNN(Polański, et al. Bioinformatics 2020) to correct for batches. As depicted in Figure R1, the results of xTrimoGene+BBKNN combination mitigated batch effects and maintained cell-type variations.
>
> > **Q4: Rare cell types**
>
> **A4**: To investigate how xTrimoGene behaves on these unseen data, we conduct the following evaluation analysis.
>
> 1. We first collected the scRNA-seq data from a previous study (Ji et al., 2020, Cell), which profiles human skin squamous cell carcinoma landscape with scRNA-seq and spatial transcriptomics technology simultaneously. The scRNA-seq data is not present during xTrimoGene training process. Notably, the authors found that clustering the scRNA-seq data (Figure R2, left panel) yields a novel cell subgroup named TSK (Tumor-Specific Keratinocyte), which is clearly in the tumor sample  but not the normal sample.
>
> 2. We employed the tumor scRNA-seq data to explore whether xTrimoGene is robust to distinguish the cell subpopulations from others. The expression matrix is fed into xTrimoGene and the dumped context embedding is used for subsequent UMAP visualization. The results show that the TSK subgroup is clearly separated (Figure R2, right panel). More importantly, the two TSK subgroups are merged with xTrimoGene, demonstrating its generalization ability to generate good cell-specific embeddings for unseen data.
>
> > **Q5: Memory/hardware requirements?**
>
> **A5**: For the pre-trained xTrimoGene models, the memory consumption for inference with a sample of approximately 2000 non-zero expressed genes is approximately 50GB for the xTrimoGene-100M model and around 18GB for the xTrimoGene-10M model. It's worth noting that, in line with our pre-training settings, we conducted our tests using bf16 mode on an Nvidia A100 80G GPU. We hope this provides clarity, and we're here to offer further details if necessary.

---

> > ### Comment · Reviewer_9yec · 2023-08-14
> >
> > Dear authors,
> >
> > Thank you for your detailed response. I just have a couple more follow-up clarification questions before making my final decision:
> >
> > **Re A3**: Are there any example results of the procedure from point (1) in the authors' response? I couldn't find anything in the rebuttal PDF or the original submission.
> >
> > **Re A4**: Could the authors provide more details on why the two TSK populations being merged in the xTrimoGene embedding space is desirable? In other words, is there a technical confounder (e.g. batch) here that xTrimoGene is removing? Given the information provided in Figure R2 it's not clear to me whether this merging behavior is good or e.g. removing distinctions between two subtypes of TSKs.

---

> > > ### Author Response · Authors · 2023-08-16
> > > **Thank you for your feedback!**
> > >
> > > **A3**: Both two strategies we discussed in the rebuttal period can remove the batches. Since the second strategy (xTrimoGene+BBKNN) only needs inferred cell embeddings and does not require any fine-tuning, we conducted an additional experiment and gave the example result (Figure R1). While the first strategy needs to fine-tune the pre-trained model and adjust the hyper-parameter setting. Due to the current discussion time limitation, we can't provide the result now, but we would like to add it in a future version. As far as the current results, the second strategy is a more lightweight approach, and it already corrected the batch effects as shown in Figure R1. Thank you again for your interest in batch correction, making us have a deeper thought on the batch correction strategies and further expand xTrimoGene for handling more downstream scenarios.
> > >
> > > **A4**: We checked the data and found a technical bias (rather than biological bias) between the two TSK sub-populations, which potentially leads to the distinctions. Specifically, we compared the total count and expressed gene number between these two groups. The results showed that the left TSK group achieves a higher sequencing quality, where the total counts are almost 1.8 times (median: 24,648 vs 13,574) over the right TSK group and the expressed gene percentage is also much higher (median: 25.3% vs 17.2%). The technical factor tends to induce the separation into two subgroups. However, the bias is removed by xTrimoGene, illustrating its efficiency in preserving biological signals.

---

> > > > ### Comment · Reviewer_9yec · 2023-08-16
> > > >
> > > > Thanks for the clarifications! My concerns have largely been addressed and (as promised) I've raised my score.

---

> > > > > ### Author Response · Authors · 2023-08-17
> > > > > **Thank you!**
> > > > >
> > > > > Thank you for your time to review our work and for your valuable feedback. We're pleased to hear that our clarifications have largely addressed your concerns. Your comments are constructive in helping us refine and strengthen our work. We greatly appreciate your efforts in evaluating the work and thank you for raising the score.

---

### Author Rebuttal · Authors · 2023-08-10

We extend our sincere gratitude to all the esteemed reviewers for dedicating their time and expertise to meticulously evaluate our work. Their valuable feedback has significantly contributed to enhancing the quality and depth of our research. We have conscientiously delved into every facet of the comments.

Guided by the insightful suggestions, we have made several noteworthy adjustments to the content. These revisions encompass elucidating intricate concepts for improved clarity, bolstering our textual content, and incorporating additional analyses. To elucidate, we have conducted a comprehensive array of ablation studies and experimental investigations, which in turn have shed light on the architectural intricacies, merits, and limitations of xTrimoGene.

Here we briefly summarize the added experiments:

1. **Ablation study on normalized value versus raw count**. The results show that pre-training with normalized gene expression value achieves a better performance than raw counts. The details are responded to Reviewer 9yec.

2. **Application on handling batch effects**. We discussed the potential usage with xTrimoGene to remove batch effects. Concretely, we provided an analysis (Figure R1) and demonstrated xTrimoGene's capability on the issue. The details are responded to Reviewer 9yec.

3. **Rare cell types detection**. We explored the ability of xTrimoGene to identify rare cell types from a large population (Figure R2). The details are responded to Reviewer 9yec.

4. **Ablation of the relative masking frequency of zeros**. Apart from the presented ablation study on masking strategy, we provided an additional investigation on zero masking frequency. The results show that the current configuration is within the optimal interval. The details are as below and responded to Reviewer MmMn.

| Value    | Masked1     | Masked2    | Masked3   | Masked4    | Masked5    | Total   |
|----------|-------------|------------|-----------|------------|------------|---------|
| $\neq$ 0 | 100(5%)    | 300(15%)  | 600(30%) | 900(45%)  | 1,100(55%) | 2,000   |
| = 0      | 1,040(5.8%) | 840(4.7%) | 540(3%)  | 240(1.3%) | 40(0.2%)  | 18,000  |
| Sum      | 1,140       | 1,140      | 1,140     | 1,140       | 1,140       | 20,000  |
| Sum ratio        | (5.7%)     | (5.7%)    | (5.7%)   | (5.7%)    | (5.7%)    | (100%) |
| ARI      | 0.6654      | 0.5043     | 0.7767    | 0.7817     | 0.5048     |         |
| NMI      | 0.7170      | 0.6884     | 0.7810    | 0.7833     | 0.6348     |         |
| HOMO     | 0.7285      | 0.7481     | 0.7841    | 0.78226    | 0.6826     |         |
| CP       | 0.7058      | 0.6375     | 0.7778    | 0.7843     | 0.5932     |         |
| SIL      | 0.1418      | 0.1522     | 0.1406    | 0.1675     | 0.1247     |         |


5. **Comparison to other models on downstream task**. We also included scBERT for comparison on the perturbation effection prediction task. The results indicate that xTrimoGene is superior over scBERT to capture the intrinsic context embedding under perturbed conditions. The details are as below and responded to Reviewer MmMn.

| Pre-trained model | Total  | 1-gene | 2-gene(seen0) | 2-gene(seen1) | 2-gene(seen2) |
|-------------------|--------|--------|---------------|---------------|---------------|
| xTrimoGene    | 0.1983 | 0.1930 | 0.2385        | 0.2100        | 0.1286        |
| scBERT            | 0.2231 | 0.2116 | 0.2581        | 0.2386        | 0.1522        |


6. **In-depth comparison with CellTypist**. For the cell type annotation task, we performed further analysis and showed advantages in two aspects: (1) xTrimoGene is more robust to identify rare cell types than CellTypist; (2) xTrimoGene embedding reveals potential cell type specific gene-gene networks (Figure R3). The details are responded to Reviewer TD4Y.

7. **Embedding visualization analysis**. We also utilized embedding visualization to interpret the behavior of three models on the cell type annotation task (Figure R4). The details are responded to Reviewer N2nG.

We take this opportunity to reiterate our gratitude to each reviewer, as their incisive comments have propelled our work. If our responses have not entirely addressed your concerns, we cordially invite you to share additional queries. We stand ready to engage in further discussions and provide any necessary clarifications.

---

> ### Comment · Reviewer_MmMn · 2023-08-11
>
> **Re 4. & 5.:**
> As mentioned in my response below, I appreciate both experiments, which, in my mind, strengthen the paper.

---

### Decision · Program_Chairs · 2023-09-21

**Decision:**

Accept (poster)

**Comment:**

All reviewers liked the paper and both authors and reviewers engaged in a good discussion that has helped improve the paper. Acceptance is recommended.